# DyVal: Dynamic Evaluation of Large Language Models for Reasoning Tasks

**Kaijie Zhu**[1][*], **Jiaao Chen**[2][*], **Jindong Wang**[1][†], **Neil Zhenqiang Gong**[3], **Diyi Yang**[4], **Xing Xie**[1]

[1]Microsoft Research, [2]Georgia Tech, [3]Duke University, [4]Stanford University

## Abstract

Large language models (LLMs) have achieved remarkable performance in various evaluation benchmarks. However, concerns are raised about potential data contamination in their considerable volume of training corpus. Moreover, the static nature and fixed complexity of current benchmarks may inadequately gauge the advancing capabilities of LLMs. In this paper, we introduce **DyVal**, a general and flexible protocol for dynamic evaluation of LLMs. Based on our framework, we build graph-informed DyVal by leveraging the structural advantage of directed acyclic graphs to dynamically generate evaluation samples with controllable complexities. DyVal generates challenging evaluation sets on reasoning tasks including mathematics, logical reasoning, and algorithm problems. We evaluate various LLMs ranging from Flan-T5-large to GPT-3.5-Turbo and GPT-4. Experiments show that LLMs perform worse in DyVal-generated evaluation samples with different complexities, highlighting the significance of dynamic evaluation. We also analyze the failure cases and results of different prompting methods. Moreover, DyVal-generated samples are not only evaluation sets, but also helpful data for fine-tuning to improve the performance of LLMs on existing benchmarks. We hope that DyVal can shed light on future evaluation research of LLMs. Code is available at: `https://github.com/microsoft/promptbench`.

## 1 Introduction

Large Language Models (LLMs) have recently achieved unprecedented performance across diverse tasks (OpenAI, 2023b; Bubeck et al., 2023). The great endeavor have led to positive speculation on the possibility of LLMs being precursors of artificial general intelligence, necessitating the creation of nuanced evaluations. By pinpointing gaps for improvements, evaluation becomes the bedrock that enhances the understanding of current models and ensures AI's continued progression.

Efforts to evaluate LLMs have become intensified significantly. Liang et al. (2023) introduced HELM, which offers a holistic assessment of LLM in various scenarios. Similarly, Chatbot Arena (Zheng et al., 2023) evaluates LLMs by contrasting their generated output. Other benchmarks that have set the standard in the realm of LLM evaluations include AlpacaEval (Li et al., 2023c), C-Eval (Huang et al., 2023), ARB (Sawada et al., 2023), API-Bank (Li et al., 2023a), Socket (Choi et al., 2023), and Big-Bench (bench authors, 2023). Moreover, manual experiments have emerged as a complementary approach to these benchmarks, with works such as Bubeck et al. (2023) and Bang et al. (2023). Complementing these, human evaluators have also been instrumental in gauging the prowess of LLMs, as discussed by Ziems et al. (2023) and Zečević et al. (2023).

Current evaluation benchmarks face two fundamental challenges. First, **data contamination.** Many benchmarks source their data from the Internet, causing potential overlap with the vast corpus on which LLMs are trained, leading to the debate of "Generalization vs. Memorization" (Bender et al., 2021; Magar & Schwartz, 2022; Carlini et al., 2023; Biderman et al., 2023): *Are the model's results stemming from genuine ability or just memorization of the training data*? A recent example is provided by Zečević et al. (2023): LLMs can ambiguously deduce the conclusion that altitude influences temperature based on seen data. Similarly, Berglund et al. (2023) found that LLMs trained

---

[*]Equal contribution. Contact: kaijiezhu11@gmail.com, jiaaochen@gatech.edu.

[†]Correspondence to: Jindong Wang <jindong.wang@microsoft.com>.

on "A is B" fail to infer "B is A", which doubts the abilities of LLMs might come from memorization. Second, **static dataset and fixed complexity.** As LLMs progress at a rapid pace, existing datasets usually fail to match the models' ever-evolving capabilities, because the *complexity* level of existing benchmarks is usually static and fixed. As Dziri et al. (2023) demonstrated, while handling simple problems pretty well, LLMs fail to solve complex problems. The inability to automatically and dynamically increase complexity levels based on existing data prevents static benchmarks from being adapted to accurately select, compare, and advance LLMs. Although there are a few existing dynamic benchmarks like DynaBench (Kiela et al., 2021) and DynaBoard (Ma et al., 2021), they rely on crowd-sourcing efforts for data collection, which might be expensive and tedious.

In this paper, we introduce **DYVAL**—a novel, general, and flexible evaluation protocol for the *dynamic* evaluation of LLMs (Sec. 3.1). The core of DYVAL is to dynamically *generate* evaluation samples on the fly instead of collecting a fixed set of data. DYVAL consists of three components: 1) the generation algorithm $\mathcal{G}$ to generate test samples with diversities; 2) the constraint $\mathcal{C}$ to modulate sample complexity and validity; and 3) the description function $\mathcal{F}$ to translate the generated samples into natural languages. Based on this framework, we propose a graph-informed DYVAL (Sec. 3.2, Figure 1) to generate data using graphs. Specifically, inspired by techniques such as the compiler principle (Alfred V et al., 2007) and parsing trees which decompose complexities (Klein & Manning, 2003; Vinyals et al., 2015), we employ directed acyclic graphs (DAG) (Thulasiraman & Swamy, 2011) to *compose* fundamental elements into more intricate problems, with each unit symbolized as a graph node. The extendable and stochastic nature of graph generation effectively regulates the complexity levels. Additionally, the hierarchical attributes of graphs suit them for multi-step inferential tasks like logics. Problems generated by DYVAL not only require profound understanding of problem solving rather than simple memorization but also echo the human approach to incremental problem-solving and solution derivation. Being general and flexible, DYVAL co-exists and co-evolves with existing benchmarks for better LLMs evaluation and evolution.

We leverage DYVAL to synthesize 7 reasoning tasks[1], encompassing: (1) Mathematics: arithmetic and linear equations; (2) Logical reasoning: boolean, deductive, and abductive logic; (3) Algorithm: reachability and maximum sum path problems. We then re-examine the state-of-the-art LLMs ranging from Flan-T5-large (Chung et al., 2022), phi-1.5 (Li et al., 2023d), Xwin-13B (Team, 2023), Llama2-13B-chat (Touvron et al., 2023), Vicuna-13B-v1.3 (Chiang et al., 2023), WizardMath-13B (Luo et al., 2023), to GPT-3.5-Turbo (OpenAI, 2023a) and GPT-4 (OpenAI, 2023b) with DY-VAL. We also test with recent prompting techniques including Few-shot (Brown et al., 2020), CoT (Wei et al., 2022), Least to Most prompting (Zhou et al., 2023b), Automatic Prompt Engineering (Zhou et al., 2023d), and Skills-in-Context prompting (Chen et al., 2023). Finally, we perform human study involving 82 human evaluators for comparison and fine-tuning experiments using DY-VAL-generated evaluation samples. Furthermore, experiments on existing benchmarks also show that fine-tuning LLMs with data generated by DYVAL could directly improve models' abilities without extra careful collection of training data (Zhou et al., 2023a). We further show the flexibility of DYVAL by extending it to natural language tasks in Appendix H. Our key findings are:

- **Results on DYVAL evaluation are not always consistent with those on existing benchmarks, indicating possible low training data quality and/or data contamination of existing LLMs** (Sec. 4.2). For instance, phi-1.5, WizardMath-13B, and Xwin-13B perform poorly on DYVAL while claiming huge improvements on existing benchmarks.
- **As difficulty increases, LLMs tend to perform worse and their performance gap becomes larger, emphasizing the lack of compositionality of current LLMs and the importance of evolving complexity evaluations** (Sec. 4.2).
- **Our error analysis based on DYVAL evaluation exhibits various failure patterns** which shed light on how to further improve LLMs. (Sec. 4.3).
- **No prompt engineering methods can perform best in all of our evaluation sets; and larger model sizes tend to achieve better performances** (Sec. 4.4).
- **DYVAL can further be utilized to generate training data to improve the abilities of LLMs.** (Sec. 5). For instance, fine-tuning the Llama2 models with our DYVAL generated data demonstrates enhanced results on 6 existing benchmarks.

---

[1]We choose reasoning tasks mainly due to (1) the intrinsic connection between reasoning proficiency and intelligence; (2) the notable progress LLMs have achieved in reasoning-centric tasks (Sawada et al., 2023). Note that DYVAL could also be applied to existing benchmarks to create new and harder evaluation data.

To sum up, this paper makes the following contributions:

- **A dynamic evaluation protocol.** DYVAL is a dynamic evaluation protocol designed to generate test samples dynamically, mitigating the issues of data contamination and static complexity.
- **A graph-informed DYVAL algorithm for evaluation of the reasoning abilities of LLMs.** We use DAGs to compose 7 reasoning problems from mathematics, logical reasoning to algorithms.
- **Extensive experiments and analysis.** We conduct extensive experiments to provide insights for evaluating and improving LLMs.

## 2 RELATED WORK

**Evaluating LLMs.** While neural networks are recognized as the universal function approximators (Cybenko, 1989) with remarkable data fitting capabilities (Zhang et al., 2021; Arpit et al., 2017), debates (Bender et al., 2021; Zhang et al., 2021; Tänzer et al., 2022; Magar & Schwartz, 2022; Carlini et al., 2023; Wu et al., 2023; Tang et al., 2023; Zečević et al., 2023; Kocoń et al., 2023; Schaeffer, 2023; Biderman et al., 2023; Zhu & Li, 2023) persist regarding the true nature of LLMs' generalization abilities. The growing prominence of LLMs necessitates rigorous benchmarks (Hendrycks et al., 2021; Li et al., 2023b; Zhong et al., 2023; HuggingFace, 2023). Recent trends include: (1) human-centric evaluations (Gao et al., 2022; Ribeiro & Lundberg, 2022), (2) crowd-sourced testing (Kiela et al., 2021; Ma et al., 2021), and (3) specialized task challenges (Liang et al., 2023; Tian et al., 2018; Ribeiro et al., 2020; bench authors, 2023). Complementing with these, our DYVAL introduces a dynamic evaluation system, consistently relevant in the swiftly evolving landscape of AI. Although Krause et al. (2018) introduced the term "dynamic evaluation", our DYVAL differs considerably in its approach and goals. Specifically, reasoning is widely recognized as the core of both human and AI. Our focus on constructing reasoning tasks mirrors the intricate and multi-step nature of human reasoning (Brody, 1999; Lohman & Lakin, 2011; Sawada et al., 2023), building reasoning benchmarks is a critical step to help LLMs towards intelligence.

**Data Contamination.** Researchers start to realize the potential data contamination problem in LLMs (Lovin, 2023; Chowdhuri et al., 2023; Bender et al., 2021; Kocoń et al., 2023). The GPT-4 and LLama reports clearly stated the phenomenon of data contamination. Recently, Zhou et al. (2023c) discussed the risks and impacts of data contamination of evaluation benchmarks in assessing LLMs. Li (2023) examined the data contamination problem of LLama models. The Skywork LLM Wei et al. (2023) again demonstrated the data contamination issue in several. Golchin & Surdeanu (2023a;b); Oren et al. (2023); Yang et al. (2023b) designed novel methods to detect the data contamination of LLMs. DYVAL is not a detection approach but a new protocol to mitigate the contamination issue.

**Complex-to-simple problem decomposition and evaluation set construction.** Employing *graphs* to deconstruct complex tasks has been an enduring and effective strategy across domains. Compilers, as seen in computational theory (Alfred V et al., 2007), effectively break down high-level constructs, while in NLP, parsing trees bring clarity to intricate syntactic and semantic structures (Klein & Manning, 2003; Vinyals et al., 2015). Roy & Roth (2015) displayed the potency of this method in arithmetic, using trees for solving multi-step problems. Additionally, several contemporary techniques have implored LLMs to decompose complex problems (Wei et al., 2022; Zhou et al., 2023b; Khot et al., 2022; Zhang et al., 2023). Several studies have leveraged graph-based approaches for constructing compositional tasks, particularly in the domains of first-order logic (Sinha et al., 2019; Clark et al., 2020; Tian et al., 2021) and causal reasoning (Jin et al., 2023). DYVAL presents notable distinctions in both objective and methodology. Additionally, GraphWorld (Palowitch et al., 2022) primarily benchmarks Graph Neural Networks (GNNs), whereas DYVAL focuses on LLMs using the graph structure. They are different in nature.

## 3 DYVAL

In this section, we first elucidate our general dynamic evaluation protocol to address the challenges of data contamination with dynamic data generation and controllable complexity in Sec. 3.1. We then adapt this general protocol for reasoning tasks by leveraging the Directed Acyclic Graphs (DAGs) in Sec. 3.2. More analysis of the flexibility of DYVAL is in Sec. 3.3.

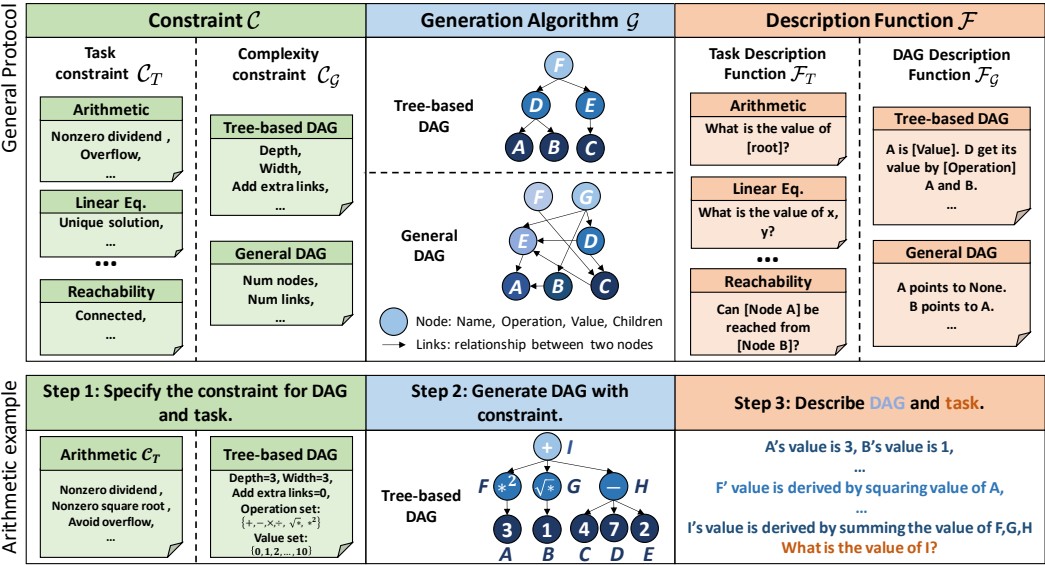

Figure 1: The pipeline of the graph-informed DYVAL. Up: the general evaluation framework; down: an arithmetic example. More details can be found at Sec. 3.2 and Appendix B.

## 3.1 GENERAL DYNAMIC EVALUATION DESCRIPTION LANGUAGE

First, we introduce the general description language of the dynamic evaluation protocol. Given a task $T$, a dynamic evaluation algorithm is formulated as $\mathcal{A}_T = \mathcal{F}(\mathcal{G}(\mathcal{C}))$, where (1) $\mathcal{G}$ is the **sample generation algorithm**, incorporating randomness to guarantee uniqueness of each sample. Randomness may vary in different tasks, such as the numbers in math problems and the logic chains in a logic reasoning task. (2) $\mathcal{C} = \{\mathcal{C}_T, \mathcal{C}_\mathcal{G}\}$ denotes **constraints** on $\mathcal{G}$, where $\mathcal{C}_T$ is the task constraint for task $T$ such as the legality guarantee of the generated samples in the context of the task. $\mathcal{C}_\mathcal{G}$ is the complexity constraint for the generation process, such as the sampling strategy for the value at each node and the number of perturbations added to the evaluation samples. (3) $\mathcal{F} = \{\mathcal{F}_T, \mathcal{F}_\mathcal{G}\}$ is the **description function** to translate the raw evaluation samples generated by $\mathcal{G}$ into natural language descriptions. $\mathcal{F}_\mathcal{G}$ elucidates the characteristics and properties of the samples generated by $\mathcal{G}$. $\mathcal{F}_T$ is the description for task $T$ such as task objective and expected outcomes.

In general, an evaluation sample can be represented as $d_{\text{eval}} = \mathcal{F}_T(\mathcal{F}_\mathcal{G}(\mathcal{G}(\mathcal{C}_\mathcal{G}, \mathcal{C}_T)))$ using the above description language. $\mathcal{G}$ first produces a sample that adheres to the complexity constraint $\mathcal{C}_\mathcal{G}$ and the task constraint $\mathcal{C}_T$. Then it undergoes transformation by description function $\mathcal{F}_\mathcal{G}$ into a natural language format and finally goes through the task description function $\mathcal{F}_T$. The description language above naturally (1) avoids data contamination by dynamic generation through $\mathcal{G}$, and (2) promises dynamic datasets and controllable complexity through $\mathcal{C}$. Specifically, by varying the constraints in $\mathcal{C}$, we can generate evaluation samples of different difficulties, allowing "co-evolution" of both the LLMs and the evaluation process. The description language is flexible since it allows for different generation algorithms and complexity control by changing $\mathcal{G}$ and $\mathcal{C}$ accordingly.

## 3.2 GRAPH-INFORMED DYNAMIC EVALUATION FOR REASONING TASKS

In this section, following the general evaluation description language, we implement DYVAL for reasoning tasks by taking inspiration from the graph structure. Given the intrinsic multistep inferential nature of reasoning tasks, they inherently exhibit structural characteristics, making directed acyclic graphs (DAGs) a natural choice for modeling these tasks. DAGs also facilitate dynamic sample generation by modulating the internal structure and fine-grained control over problem difficulty by adjusting the structural complexity. More background of DAGs can be found in Appendix A.

### 3.2.1 GENERATION ALGORITHM $\mathcal{G}$: DAG CONSTRUCTION

The generation algorithm is established on the graph construction process. We categorize DAGs as Tree-based DAGs (T-DAGs) and General DAGs (G-DAGs), illustrated in Figure 1. T-DAGs are inherently hierarchical, making them suitable for tasks that proceed from a set of initial premises to

Table 1: Three types of reasoning tasks generated by DYVAL.

| Field | Task | Generation algorithm $\mathcal{G}$ | Constraint $\mathcal{C}$ | | # Classes | Description $\mathcal{F}$ |
|---|---|---|---|---|---|---|
| | | | $\mathcal{C}_T$ | $\mathcal{C}_{\mathcal{G}}$ | | |
| Mathematics | Arithmetic | Tree-based | $\mathcal{V} : \{1, 2, \ldots, 10\}$ $\mathcal{O} : \{+, -, \times, \sqrt{\cdot}, \cdot^2\}$ | Depth, Width, Extra links, Random desc | - | `What is the value of [Root]?` |
| | Linear equation | Tree-based | $\mathcal{V} : \{1, 2, \ldots, 10\}$ $\mathcal{O} : \{+, -, \times, \sqrt{\cdot}, \cdot^2\}$ | Depth, Width, Extra links, Random desc | - | `What is the value of x and y?` |
| Logical Reasoning | Bool | Tree-based | $\mathcal{V} : \{\text{True}, \text{False}\}$ $\mathcal{O} : \{\text{AND}, \text{OR}, \text{NOT}\}$ | Depth, Width, Extra links, Random desc | 2 {True, False} | `What is the value of [Root]?` |
| | Deductive | Tree-based | $\mathcal{V} : \{\text{True}, \text{False}\}$ $\mathcal{O} : \{\text{AND}, \text{OR}, \text{NOT}\}$ | Depth, Width, Extra links, Random desc | 3 {True, False, N/A} | `What is the value of [Root]?` |
| | Abductive | Tree-based | $\mathcal{V} : \{\text{True}, \text{False}\}$ $\mathcal{O} : \{\text{AND}, \text{OR}, \text{NOT}\}$ | Depth, Width, Extra links, Random desc | 3 {True, False, N/A} | `Given [Root] is [Value], what is the value of [Leaf i]?` |
| Algorithm | Reachability | General | $\mathcal{V} : -$ $\mathcal{O} : -$ | # Nodes, # max links, random desc | 2 {True, False} | `Can [Node i] be reached from [Node j]?` |
| | Max sum path | General | $\mathcal{V} : \{1, 2, \ldots, 10\}$ $\mathcal{O} : -$ | # Nodes, # max links, random desc | - | `What is the maximum path [Node i] to [Node j]?` |

a final inference, such as arithmetic problems and logical reasoning tasks. Each node in T-DAGs represents a foundational subproblem. These subproblems are chained by the links between nodes and finally form a complex problem. Conversely, G-DAGs excel in mapping intricate relationships, especially in tasks demanding understanding of non-linear interactions. They are ideal for algorithmic challenges involving complex dependencies. For instance, imagine modeling a system where a change in one entity might impact multiple others in a cascading fashion, or tasks require finding different potential pathways between entities. The generation process for these two types of DAGs are presented in Appendix B.1.

**Randomness in DAGs generation process.** T-DAG randomness arises from operations assigned to the nodes and the initial values of the leaf nodes. For instance, in arithmetic, the operation can be "+", with the leaf nodes receiving random numbers. On the other hand, for G-DAGs, each node is endowed with a random value (if needed for a certain problem). For every node, the number of children is determined randomly, and the maximum number of children depends on the input. We then establish the links by selecting the target child nodes at random.

Theorems 3.1 and 3.2 formally guarantee the dynamic generation process by exploring the probability that two samples generated by T-DAG and G-DAG are identical. We focus exclusively on the base case, setting aside additional complexities like the integration of random links or the embedding of random descriptions, which would further diminish the likelihood of two DAGs being identical.

**Theorem 3.1.** Given a tree-based DAG with depth $d$ and width $w$, if the operation set for non-leaf nodes has $k$ distinct operations and the value set for leaf nodes contains $n$ distinct values, the probability that two independently generated DAGs are identical is: $P = \left( k^{\frac{w^{d-1}-1}{w-1}} \times n^{w^{d-1}} \right)^{-1}$.

**Theorem 3.2.** Given a general DAG with $n$ nodes where each node has a minimum of $l \geq 1$ links, the probability that two randomly selected DAGs are identical is bounded by $\frac{1}{(n-1)!}$.

Proofs can be found in Appendix C. These theorems guarantee that the odds of producing identical evaluation samples are considerably low. For instance, in the arithmetic task (where $k = 6, n = 10$) with $d = 4$ and $w = 2$, the chances that two DAGs are identical hover around $1e^{-15}$.

### 3.2.2 CONSTRAINTS $\mathcal{C}$ FOR GRAPH GENERATION

**Task constraint $\mathcal{C}_T$.** Task constraints vary for tasks. Take the node creation for instance: 1) What distribution should the node value adhere to? 2) What set of operations is permissible? 3) How should a node's value be computed from its children's values? In arithmetic tasks, $\mathcal{C}_T$ includes ensuring that a dividend is nonzero, avoiding overflow, etc. Here, we concentrate on two general task constraints: (1) *Value distribution $\mathcal{V}$:* Specifies the permissible range or distribution from which leaf node values can be assigned. For example, in logic reasoning tasks, the premises (leaf nodes) are assigned either as True or False. (2) *Operation set $\mathcal{O}$:* Lists the operations allowed within the DAG. The operation set constraint is usually used for tree-based DAGs. For example, in an arithmetic task, the set of allowed operations can be defined as the basic arithmetic operations $\{+, -, \times, /\}$.

**Complexity constraint $\mathcal{C}_{\mathcal{G}}$.** We investigate 4 techniques to inject complexity into DAGs (Figure 5): (1) *Change width and depth for T-DAGs:* The natural way to control tree complexity. (2) *Change number of nodes and links for G-DAGs:* We control the total number of nodes in G-DAGs. The

number of links in each node is randomly selected from a predefined range, e.g., $[1, 5]$. (3) *Add extra random links:* For each node, we may introduce an additional link to another random node. (4) *Embed random descriptions:* Add random descriptions to the primary DAG's descriptions. More details of complexity can be found in Appendix B.2 with Figure 7 as illustrations.

### 3.2.3 DESCRIPTION FUNCTION $\mathcal{F}$

After constructing DAGs with certain constraints, we then need to convert them into comprehensible natural language descriptions using the description function $\mathcal{F}$.

**DAG description function $\mathcal{F}_{\mathcal{G}}$.** We describe the DAG node by node and then form the description of the nodes into sequences. The interpretation of each node in natural language depends on its position and the task. For leaf nodes that represent primary input or premises, they can be described as: "`The value of [Name] is [Value].`" For instance, a node denoting number 5 could be expressed as: "`The value of node A is 5.`" For T-DAGs, the intermediate nodes that typically denote operations performed on their child nodes, the description can be formulated as: "`The value of [Name] is derived by [Operation] the values of [Children's Names].`" For G-DAG, the intermediate nodes are usually described as the connections between nodes: "`The [Name] points to [Children's Names]`". Note that natural language descriptions can be replaced according to custom needs and can be further incorporated with textual adversarial attacks (Li et al., 2019; Gao et al., 2018; Jin et al., 2020; Li et al., 2020).

Moreover, complexity is also influenced by the *order* that nodes are described. We design three orders: *topological*, *reversed topological*, and *random* orders, each offering a unique challenge in understanding the DAGs. The details of these orders are presented in Appendix B.4.

**Task description function $\mathcal{F}_T$.** The construction of $\mathcal{F}$ highly depends on the context of tasks. Notably, this construction is also highly flexible. For instance, incorporating adversarial prompts (Zhu et al., 2023) to the task description can make problems more difficult. Here we present the task description function for arithmetic and reachability tasks that are representative of T-DAG and G-DAG, respectively. Appendix B.3 presents details and examples of the remaining 5 tasks.

*Arithmetic:* Given a T-DAG, the DAG description function has already demonstrated the premise: the leaf nodes and the intermediate steps of inference: non-leaf nodes. Next, we select the root node as the variable required to solve, we append the question "`What is the value of [Root]?`" to the description where `[Root]` is filled with the name of the root variable (Figure 8).

*Reachability:* The reachability task aims to model if two nodes are connected in a graph. For a G-DAG, the DAG description function has demonstrated the connections between nodes. The task description for reachability task is: "`Can the [Node i] be reached by [Node j]`" where `Node` $i$ and `Node` $j$ are randomly selected from the nodes in G-DAG (Figure 9).

Finally, while it is feasible to directly adopt GPT-4 to generate a contextualized description rather than the plain one (see Appendix B.5), it is challenging to verify the rationale of the problems generated by GPT-4. Thus, we leave it for future work.

### 3.3 DYVAL COEXISTS AND CO-EVOLVES WITH EXISTING BENCHMARKS

DYVAL is complementary to existing benchmarks. First, tasks with an intrinsic structure benefit significantly from DYVAL since they can modulate complexity and randomness by adjusting the generation process. Efforts such as CheckList (Ribeiro et al., 2020), data augmentation (Andreas, 2020; Zhang et al., 2022), and reasoning dataset synthesis (Sinha et al., 2019; Zhao et al., 2019; Clark et al., 2020; Tian et al., 2021; Jin et al., 2023) can be easily integrated into DYVAL. On the contrary, tasks without a well-defined structure may present challenges for DYVAL's implementation. Second, DYVAL can be enhanced by existing benchmarks to formulate more challenging scenarios. For instance, the description function $\mathcal{F}$ is all about natural language texts, so it can be easily combined with adversarial attacks (Li et al., 2019; Jin et al., 2020; Zhu et al., 2023) or out-of-distribution prompts (Yang et al., 2023a) to assess the robustness of LLMs.

Note that while this papers focuses on evaluating reasoning tasks, DYVAL is *flexible* to evaluate natural language tasks. We show an initial study using DYVAL to evaluate sentiment analysis in Appendix H and more work can be done in the future. Finally, DYVAL guarantees an unbiased and

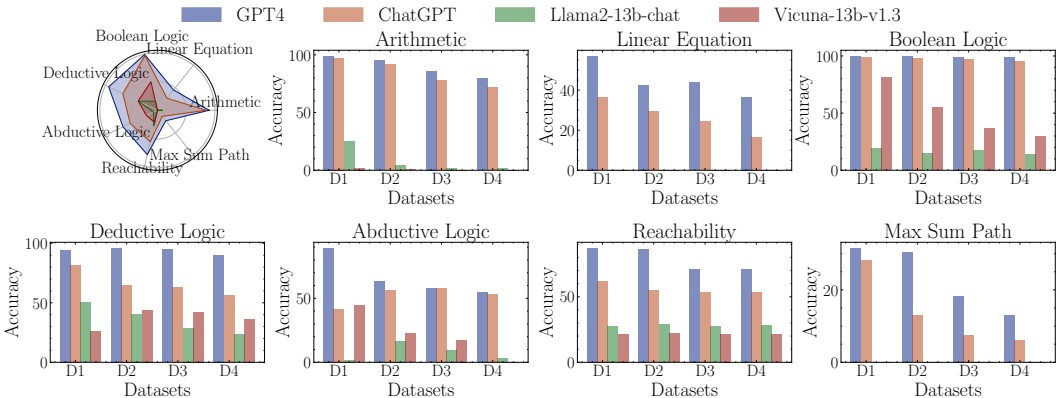

Figure 2: Results on 7 tasks with complexity from D1 to D4 (averaged on 3 description orders and 3 seeds). Xwin-13B, phi-1.5, and WizardMath-13B are not shown as their results are all 0.

balanced construction of evaluation samples by nature, since one can easily control the generation process, as shown in Appendix F.

# 4 EXPERIMENT

## 4.1 SETUP

**Tasks and complexity level.** We mainly discuss the constraint used in each task. Test set accuracy might differ as it is generated dynamically. To balance test time and discrepancy, we produce 500 samples for each dataset. To mitigate the impact of randomness on evaluation results, we assess each dataset three times. We define 4 complexity levels (D1∼D4) for each task. For tasks that use general DAGs, the number of nodes is set to $\{7, 10, 15, 20\}$ with each node having $\{3, 4, 6, 8\}$ maximum links and 1 minimum link. For tasks that use tree-based DAGs, tree depths and widths are $(2, 2), (3, 2), (3, 3), (4, 2)$, respectively. More details of D1∼D4 are presented in Appendix D.

**Evaluation metric.** Our primary evaluation metric is accuracy. For tasks where answers are numerical, we employ relative precision (Burden et al., 2015) to determine the correctness of a prediction, i.e., an answer is deemed correct if its relative precision is within a specified threshold, $\sigma$ (e.g., $0.01\%$), in relation to the ground truth value. Relative precision is calculated as $|\text{pred} - \text{gt}|/(\text{gt} + \epsilon) \leq \sigma$ where gt represents the ground truth value, pred is the model's prediction, $|\cdot|$ is the absolute value function, $\sigma$ is the desired relative precision threshold, and $\epsilon$ is a small value introduced to prevent division by zero.

**LLMs.** Our evaluated LLMs include Flan-T5-large (Chung et al., 2022), phi-1.5 (Li et al., 2023d), WizardMath-13B (Luo et al., 2023), Xwin-13B (Team, 2023), Llama2-13B-chat (Touvron et al., 2023), Vicuna-13B-v1.3 (Chiang et al., 2023), GPT-3.5-Turbo (OpenAI, 2023a), and GPT-4 (OpenAI, 2023b). Temperature is set to 0 to avoid randomness. We set the generation length to be directly proportional to the input length. Specifically, for GPT-3.5-Turbo and GPT-4, the generate length is set to be twice the input length; for the remaining models, it is set to be five times the input length. We designed prompts for each task, incorporating demonstrations of rules, particularly for reasoning and algorithm tasks. To ensure formatted output, we further ask LLMs to explicitly output their predictions between "⟨⟨⟨" and "⟩⟩⟩". All implementations are based on Huggingface.

## 4.2 RESULTS FOR MATH, LOGICAL REASONING, AND ALGORITHM TASKS

Before presenting the main results, note that **the results of Flan-T5-large, phi-1.5, WizardMath-13B, and Xwin-13B in all tasks are 0**, so we no longer report them. We carried out experiments using three random seeds. Figure 2 shows the results of all tasks averaged in three generation orders and three random seeds (full results in Appendix D.4). GPT-4 performs best, followed closely by GPT-3.5-Turbo. Llama2-13B-chat's performance is subpar, with Vicuna-13B-v1.3 occasionally outperforming Llama2-13b-chat. More findings are as follows.

**Inconsistent performance between existing static benchmarks and DYVAL:** Despite the excellent results of phi-1.5, Xwin-13B and WizardMath-13B on existing benchmarks, their poor performance in our evaluations highlights the potential problems when evaluating LLMs solely on static benchmarks and possible low training data quality or data contamination issue.

**Difficulty with complex datasets:** Performance mostly decreases sharply from D1 to D4, highlighting LLMs' struggles with increasing complexity. For example, GPT-3.5-Turbo's performance drops by 23% for arithmetic task as complexity increases. Notably, performance in abductive logic (inferring premises from conclusions) is much lower than in deductive logic (deriving conclusions from premises), as supported by Berglund et al. (2023), which shows LLMs excel more in "A is B" than "B is A". In addition, the performance difference between GPT-4 and GPT-3.5-Turbo, while subtle in simpler tasks like D1, becomes prominent in complex tasks. These observations indicate the value of intricate and evolving tasks to effectively differentiate and evaluate models. We also present more interesting observations in Appendix D.4.

**Human study:** We recruited 82 human evaluators with at least a bachelor's degree[2], to gauge their skills against LLMs on the most complex dataset (D4) for mathematical and logical reasoning tasks. Each participant tackled 5 problems from each dataset. As depicted in Figure 3, both GPT-4 and GPT-3.5-Turbo consistently showed high competence in most tasks, surpassing average human results. The reason could be that the generated problems are generally harder for humans but easier for LLMs. Nevertheless, GPT-4 struggled in areas like linear equations and abductive logic. This indicates that future development could involve more data from specific domains.

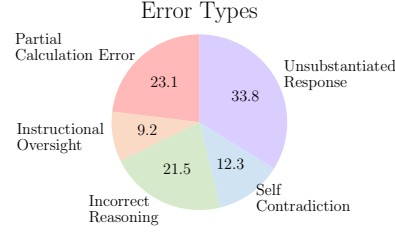

Figure 3: Human vs. LLMs results.

### 4.3 CASE STUDY

In an endeavor to comprehensively understand the behavior of LLMs, we meticulously examined the failure modes. Our focus is especially on the most challenging datasets for arithmetic, deductive logic, abductive logic, and reachability tasks based on the performance of GPT-4. We randomly selected 20 failure samples for each task and summarized the failure modes in Figure 4. The detailed failure cases are presented in Appendix D.5. The error types vary, indicating that there is much room for improvement.

Figure 4: Failure modes distribution.

**Partial calculation error:** GPT-4 occasionally errs in intermediate steps, while keeping the remaining steps correct. We emphasize that the errors may be as simple as $20/7 = 37.28$. This aligns with (Dziri et al., 2023) noting LLMs sometimes give partially correct multi-digit multiplication results. **Incorrect reasoning and self contradiction:** In reasoning tasks, GPT-4 may misinterpret rules. Given an abductive logic $A \vee B \rightarrow C$ with $C$ is False, the premise $A, B$ must be False. However, GPT-4 inaccurately abduced that either A or B *might* be False. Further, GPT-4 occasionally contradicts itself in its assumptions for the same inference in abductive logic task. **Unsubstantiated response:** In reasoning tasks and algorithm tasks, GPT-4 often answers without any inferences or justifications. Its answer-only responses suggest possible memorization or shallow understanding. **Instructional oversight:** Occasionally, GPT-4 adeptly arrives at the correct computation but stumbles when it comes to adhering to the output instructions laid out in prompts, for example, the required relative precision of mathematic calculation.

### 4.4 ABLATION STUDY

**Impact of complexity constraints** $\mathcal{C}_\mathcal{G}$**:** In Figure 5, we vary complexity in GPT-3.5-Turbo by adjusting constraints as described in Sec. 3.2.2 and observe how LLMs performance shifts across arithmetic, boolean logic, and deductive logic tasks. Notably, as task intricacy rises due to augmented complexity parameters, LLMs' performance diminishes. Depth emerges as the predominant challenge in tree-based DAGs, emphasizing the LLMs' difficulty with extended inference steps.

**Prompt engineering:** We evaluate five prompting techniques (PE) on our most challenging datasets, as outlined in Table 5 and Appendix D.7. No PE methods can perform best in all tasks. While APE notably boosts the Linear Equation task by 10%, it negatively impacts deductive and abductive logic. These varied outcomes highlight the importance of task-specific PE selection and development.

---

[2]The results may not represent the highest level of human performance. Demographics are in Appendix D.8.

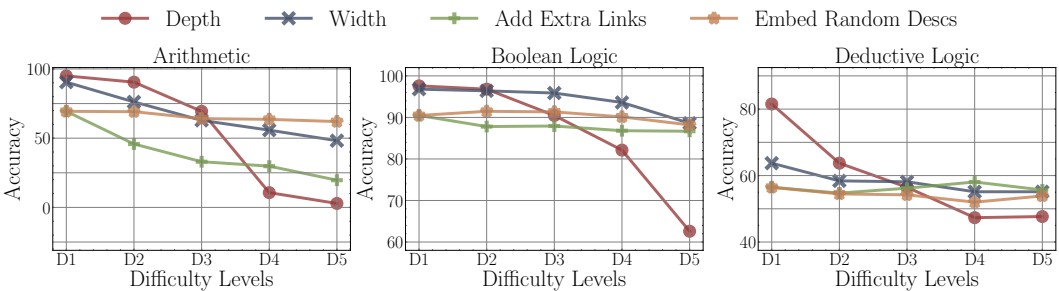

Figure 5: Comparison results across different complexity constraints.

**Influence of model size:** We further evaluate the performance of Llama2 with different model sizes of arithmetic, boolean logic and reachability tasks on their simplest dataset D1. Table 6 shows that larger sizes produce better results, but mostly still not surpass GPT-4 and human.

## 5  DYVAL HELPS FINE-TUNING

In this section, we show that DYVAL-generated data can further be utilized to fine-tune LLMs to improve their capabilities of solving complex tasks. Specifically, we generate training data for 7 tasks to fine-tune Llama2-13B-chat. The details of fine-tuning and training sample generation are in Appendix E. We then test the model with different settings: (1) *in-distribution* samples with the same difficulty as the training data; (2) *out-of-distribution* samples, whose difficulty levels are higher than the training data. To further demonstrate the effectiveness of our generated data, we test the models with few-shot examples on **existing benchmarks** including GSM8K (Cobbe et al., 2021) and SVAMP (Patel et al., 2021) to evaluate math abilities, FOLIO (Han et al., 2022) and RACO (bench authors, 2023) to evaluate the logical reasoning abilities, and DP (Dziri et al., 2023) and LCS (bench

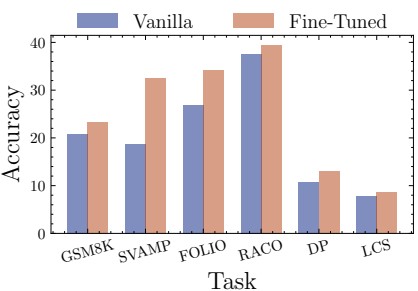

Figure 6: Results on existing benchmarks using Llama2-13B-chat model fine-tuned on DYVAL-generated data.

authors, 2023) to evaluate the algorithm abilities. Results in Figure 6 and 10 show that the performance of the fine-tuned model increases in all tasks. It shows that DYVAL is effective not only as a benchmark but also in enhancing the performance of LLMs on existing benchmarks via fine-tuning on its generated samples. The improvement might stem from the similarities between various benchmarks and DYVAL-generated samples. For instance, GSM8K samples can be interpreted as trees of depth 2 or 3. Interestingly, even no dynamic programming tasks in our fine-tuning, the fine-tuned model also showed improved performance on the DP and LCS datasets. This underscores the potential learning capability of LLMs and the efficacy of training samples generated by DYVAL. We further fine-tuned GPT-3.5-Turbo and examined its ability on general natural language understanding. The results indicated that fine-tuning on our generated datasets does not necessarily hurt the natural language understanding ability, as comprehensively discussed in Appendix G.

## 6  CONCLUSION AND DISCUSSION

We proposed DYVAL, a dynamic LLMs evaluation protocol to mitigate the data contamination and static complexity of existing benchmarks. We designed the graph-informed DYVAL for reasoning tasks. The strength of DYVAL lies in its dynamic generation of samples, with inherent flexibility for difficulty adjustment. We observed several interesting findings in experiments using our benchmark. More importantly, DYVAL-generated samples can not only be used as evaluation samples, but also act as fine-tuning data for LLMs to enhance their performance in existing benchmarks.

Our work has several limitations. (1) Tasks: We currently focused on reasoning tasks. While DYVAL supports other tasks (see Sec. H), it requires design of the generation algorithm $\mathcal{G}$. (2) Samples: Our experiments utilized a limited set of test samples due to resource constraints. Evaluations on larger sets may help to observe more findings. (3) Fine-tuning: Fine-tuning can be done on more diverse models and datasets to gain deeper insights.

## ACKNOWLEDGEMENT AND DISCLAIMER

The purpose of this research is to present a dynamic and evolving evaluation protocol in response to the rapid development of LLMs. We have the following claims. First, the generation mechanism of DYVAL does not contain any potentially harmful words or expressions but only mathematical, logical, and algorithmic descriptions. In the future, the usage of DYVAL on other natural language tasks should be dealt with cautions to not include any harmful or irresponsible languages. Second, human subjects are involved in this study to act as LLMs' competitors for performance comparison and analysis. All human studies are conducted obeying laws and regulations in certain countries. Third, the experiments on GPT-3.5-Turbo and GPT-4 conducted in this paper are based on their latest version in June, 2023. Authors recommend using the same version of these services for reproducibility. As we tried our best to tune the best prompts for our experiments, it is, however, well-known that LLMs are highly sensitive to prompts. Therefore, the experiments in this paper are only based on our prompt design and codebase. Finally, we may have concluded that some LLMs in this paper achieved poor performance in our benchmark, but this does not mean these models are not good or cannot be used in practice. Authors remain positive and optimistic to all evaluated LLMs that they will further be stronger.

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

CONTENTS

A PRELIMINARY ON DIRECTED ACYCLIC GRAPH

Directed Acyclic Graphs, commonly referred to as DAGs, are a category of graphs that encapsulate a unique structure: they are directed and contain no cycles. In a DAG, vertices are connected by directed links, and there exists no sequence of links that loops back to an original node. Every link in a DAG has an initial node and a terminal node, giving it a direction. This is often symbolized as $a \rightarrow b$, where a is the starting node and b is the ending node. A key property that differentiates DAGs from other directed graphs is their lack of cycles. In other words, starting from any node in the graph, one cannot traverse a series of links and return to the same node.

In our implementation, each *node* comprises three attributes: 1) *Children (Links)*: These are the direct dependents or subsequent nodes that a given node connects to. They highlight the immediate

relations or following a particular node. 2) *Value*: Every node possesses a value, which can either be explicitly assigned or derived based on its operation and its children. This value captures the essence or result of the represented subproblem. 3) *Operation*: Especially relevant in tree-based DAGs, the operation dictates how a node interprets or processes the values of its children to compute its own value. Operations might include mathematical functions, logical evaluations.

## B  DETAILS OF DYVAL

### B.1  GENERATION ALGORITHM

We distinguish DAGs into two primary categories: Tree-based DAGs (T-DAGs) and General DAGs (G-DAGs), as shown in Figure 1.

#### B.1.1  T-DAGS

Tree-based DAGs possess an innate hierarchical structure that frequently encapsulate tasks that entail a sequence of conditions culminating in a definitive conclusion or result. This hierarchy naturally aligns with the structure of many mathematical and logical problems. For instance, in solving a multi-step algebraic problem, one often starts with the provided equations (leaf nodes) and proceeds step-by-step, combining and reducing these equations until arriving at the final solution (root node). Such a natural progression of deduction makes tree-based DAGs particularly feasible for these problems.

We employ a top-down approach to construct a Tree-based DAG. This algorithm is tailored to produce a tree with a designated depth and width. The inherent randomness stems from two main factors: the operations assigned to intermediate nodes and the initialization values of the leaf nodes. For the intermediate nodes, we commence by randomly selecting an operation that defines the relationship between the node and its children. Take an arithmetic task as an example: selecting 'addition (+)' implies that the node's value is the sum of its children's values. Once all child nodes are established, we compute the parent node's value accordingly. In the case of leaf nodes, values are assigned randomly, such as picking an integer from the range $[1, 10]$ for arithmetic tasks.

#### B.1.2  G-DAGS

General DAGs, diverging from tree-based ones, lack a strict hierarchy. Instead, they present a more intricate web of node relationships. Their strength lies in simulating complex, intertwined relations in real-world situations. A classic use-case is the representation of transportation systems where nodes symbolize cities and edges represent connecting roads. Challenges such as determining if one city is accessible from another encapsulate the real-world problems general DAGs adeptly model. Their flexibility extends to representing a plethora of situations, from mapping supply-chain logistics to analyzing social networks.

To create a general DAG, we initiate by generating isolated nodes without any connecting links. Subsequently, each node is endowed with a random value. For every node, the number of children is determined randomly, the maximum number of children is depended on the input. We then establish the links by selecting target child nodes at random.

### B.2  COMPLEXITY CONTROL

Figure 7 demonstrated 4 types of complexity constraints for T-DAGs. Compared to original case, adding width and additional links augments the computational intricacy of each subproblem. Increasing the depth escalates the complexity by necessitating more inference steps. Embedding random descriptions aims to distract LLMs.

### B.3  DESCRIPTION FUNCTION

Figure 9 presented an illustration of our generated 7 tasks in 3 subjects: (1) Mathematics (DYVAL-M), which includes arithmetic task and linera equation task; (2) Logical Reasoning (DYVAL-L),

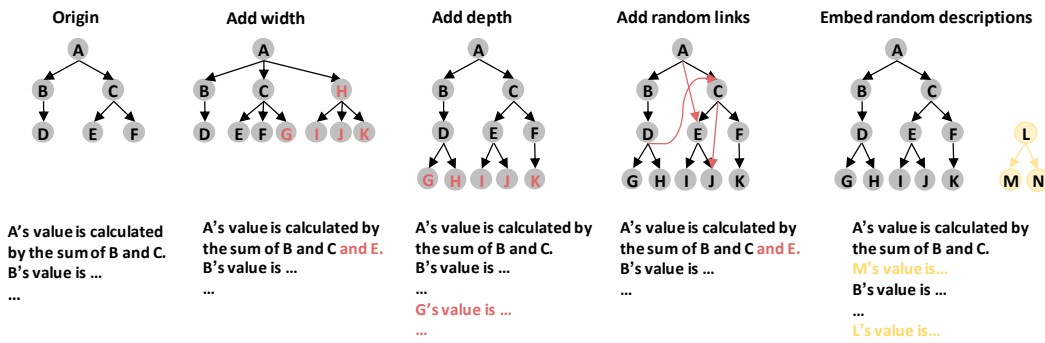

Figure 7: The complexity constraints for Tree-based DAGs.

which includes boolean logic task, deductive logic task, and abductive logic task; (3) Algorithm Tasks (DYVAL-A), which includes reachability task and max sum path task.

### B.3.1 DYVAL-M

For DYVAL-M, we design mathematical problems that can be categorized into two main types:

**Arithmetic:** Given a T-DAG, the DAG description function has already demonstrated the premise: the leaf nodes and the intermediate steps of inference: non-leaf nodes. Next, we select the root node as the the variable required to solve, we append the question "What is the value of [Root]?" to the final of the description where [Root] is filled with the name of the root variable.

```
Here is a description of an arithmetic problem:

The value of aaa is 9.

The value of aad is 4.

aae gets its value by taking the square root of the value that aad
has.

The value of aab is 3.

aac gets its value by adding together the value of aaa and aab.

aaf gets its value by subtracting the value of aae from the value of
aac.

Compute the result of aaf.  If the solution cannot be calculated,
answer 'N/A'. Ensure your result is within a relative precision of
0.0001 (or 0.01%) compared to the ground truth value.  Ensure your
final result begins with '<<<' and ends with '>>>', for example, if
the answer is 1, your final result should be <<<1>>>.
```

**Linear Equations:** Linear equations with multiple variables present a higher degree of complexity compared to arithmetic. We use two-variable linear equations described as $a_1x + b_1y = c_1, a_2x + b_2y = c_2$. The coefficients are assigned a random value. We ask LLMs to solve the value of $x, y$ for this linear system. Note that constructing such linear equations does not need T-DAGs or G-DAGs. To introduce additional challenges, some coefficients can be substituted with values derived from the T-DAG's roots, forcing a two-step problem-solving approach: first calculating the coefficients from the DAG and subsequently resolving the linear equations. Note that in our experiment, the tree depth and width for linear equation task are $(1, 1), (2, 2), (3, 2), (4, 2)$ respectively. $(1, 1)$ represent that the value of the replaced coefficient is directly given.

```
Given the following linear equation system with two variables:

-7 x + aac0 y = 1

8 x + -1 y = 10

The calculation of aac0 is defined as:

The value of aab0 is 4.

The value of aaa0 is 9.

aac0 gets its value by adding together the value of aaa0 and aab0.

Determine the values of x and y.  Ensure your results are within a
relative precision of 0.001 (or 0.1%) compared to the ground truth
values.  Your response should be formatted as:  <<<x's value y's
value>>>, e.g., if x=1 and y=2, then it should be <<<1 2>>>
```

### B.3.2 DYVAL-L

DYVAL-L also shares a natural compatibility with the structured representation of T-DAGs due to the innate progression and dependencies inherent in logical constructs. The tasks are:

**Boolean Logic:** Similar to arithmetic task, it primarily revolves around the manipulation and combination of True and False values using operators: AND, OR, NOT. The problems are presented as: `What is the truth value of [Root]?`.

```
Here is a description of a boolean logic problem:

aaa is True.

The value of aab equals to (NOT aaa).

The value of aac equals to (NOT aab).

Compute the result of aac.  If the solution can not be calculated,
answer 'N/A'. Ensure your final result begins with '<<<' and ends with
'>>>', for example, if the answer is True, your final result should be
<<<True>>>.
```

**Deductive Logic:** The process of deductive logic is similar to boolean logic, but deduction introduces a bit complexity compared to boolean logic inference. For instance, given premises $A$ (True) and $B$ (False), and the relationship $(A \land B) \rightarrow C$, the value of conclusion $C$ remains undetermined because the conjunction $(A \land B)$ is false. Given the description of T-DAGs, the problem is formulated as `By the rule of deduction, what is the value of [Root]?`

```
Here is a description of a deductive logic problem:

aab is True.

aaa is True.

(aaa and aab) -> aac.

aad is False.

(NOT aad) -> aae.

(aac or aae) -> aaf.

The symbol '->' represents a deductive relationship, e.g., A -> B
implies that if A is true, then B is true.  If A is false, B's truth
value remains undetermined (N/A). Deduct the result of aaf.  If the
```

```
solution can not be abducted, answer 'N/A'. Ensure your final result
begins with '<<<' and ends with '>>>', for example, if the answer is
True, your final result should be <<<True>>>.
```

**Abductive Logic:** It aims to hypothesize the most likely cause or explanation based on observed outcomes. When working with a T-DAG, we assign a random value to the root node. Then, we randomly select a leaf node, the problem is to determine the leaf node' value based on the given the DAG structure and root's value. The task description is `Given the value of [Root] is [value], what is the value of [Node]?`

```
Here is a description of an abductive logic problem:

(aaa or aab) -> aac.

(NOT aac) -> aad.

Given aad is False, what is the value of aab?

The symbol '->' represents a deductive relationship, e.g., A -> B
implies that if B is false, then A is false. If B is true, A's truth
value remains undetermined (N/A). If the solution can not be deducted,
answer 'N/A'. Ensure your final result begins with '<<<' and ends with
'>>>', for example, if the answer is True, your final result should be
<<<True>>>.
```

### B.3.3 DYVAL-A

DYVAL-A tasks is suitable for D-DAG since they aim to model the real-world applications. Among many problems that can be abstracted and modeled as a G-DAG, here we select two representative tasks.

**Reachability:** A classic example of where G-DAGs shine is in modeling problems like the reachability of two nodes in the DAG. Given various nodes representing cities and links indicating roads between them, the question models can help deduce the if there exists a route from one city to another. Thus, the description for this task is: "Can the [Node1] be reached by [Node2]" where Node1 and Node2 are randomly selected from the nodes in G-DAG.

```
Given a directed graph:

aai points to:  (None).

aac points to:  (aai).

aaj points to:  (aai).

aah points to:  (aai, aac, aaj).

aag points to:  (aac).

aaf points to:  (aag, aah, aaj).

aab points to:  (aaf, aah).

aaa points to:  (aag, aah, aaf, aaj).

aae points to:  (aai, aac, aaa).

aad points to:  (aab, aaf, aae).

Can aaf be reached starting from aag?

Respond with either '<<<True>>>' if reachable, or '<<<False>>>'
otherwise.
```

**Max sum path:** Compared to reachiability problem, max sum path is more complex. This problem assign a value for each city, and then requires to find a path from two cities that the sum of the values the path go through is maximum. It requires the LLMs to find all the path between two nodes, and then determine the path with maximum value. The description for this task is `What is the max sum path from [Node 1] to [Node 2]?`

```
Given a directed graph with values assigned to each node:
aaj points to:  (None).
aah points to:  (None).
aai points to:  (aah).
aag points to:  (aai).
aac points to:  (aag).
aab points to:  (aac, aag).
aaf points to:  (aai).
aae points to:  (aac, aah).
aad points to:  (aag, aae, aaj).
aaa points to:  (aae, aai, aaj, aad).
The value of aaj is 9
The value of aab is 8
The value of aah is 3
The value of aaf is 3
The value of aai is 3
The value of aae is 3
The value of aad is 6
The value of aac is 4
The value of aag is 8
The value of aaa is 4
What's the maximum sum path from aaa to aae?  For exmaple, the value
of the path A->B->C is obtained by summing the values of nodes A, B,
and C. Please format your response as <<<Answer>>>.  For example, if
the answer is 1, it should be presented as <<<1>>>.
```

## B.4 DESCRIPTION ORDER

- **Topological Order:** This approach sequences the description of nodes in a manner where every node is introduced after all its descendent nodes. Such a sequence ensures that leaf nodes are laid out prior to any operation that utilizes them (e.g., an addition or a logical AND).

- **Reversed Topological Order:** Taking an almost counter-intuitive approach, this order starts by spotlighting the culminating nodes or outcomes. Once these results are laid bare, the narrative retraces its steps, navigating backwards to the root nodes or primary inputs.

- **Random Order:** This unstructured method presents nodes in a random sequence, irrespective of their dependencies within the DAG. Such a disordered narrative challenges LLMs to independently connect the dots and derive patterns.

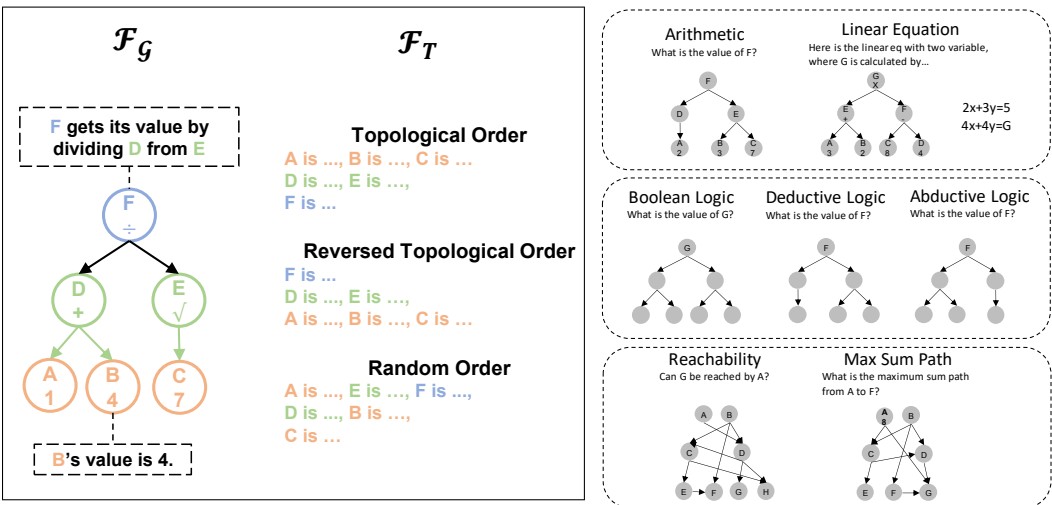

Figure 8: Description function of arithmetic task.    Figure 9: Descriptions of 7 tasks.

### B.5  POTENTIALS IN USING GPT-4 TO GENERATE DESCRIPTION FUNCTIONS

In this section, we explore the potentials in using GPT-4 to generate more natural and contextualized problem descriptions rather than the plain ones in Section B.3.

We simply feed the following instruction to GPT-4:

"`Here is a description of an arithmetic problem, please generate an application math test using the following information. Do not change the exact numbers. Note that the test should be contextualized, real applications test, instead of the plain description.`" followed by the DYVAL-generated problems such as the one in DYVAL-M.

Then, GPT-4 will generate the following application problem which is more vivid:

```
Problem:

Imagine you are managing a small warehouse.  The warehouse has a total
storage capacity of 9 large crates (represented by 'aaa').  You also
have an additional area that can store 3 more crates (represented by
'aab').

Recently, you purchased a set of 4 special crates (represented by
'aad') that are stackable.  When stacked, these crates take up the
space of 2 regular crates.

Your task is to calculate the new total storage capacity of your
warehouse after adding the stackable crates and utilizing the
additional storage area.

Steps:

Identify the total initial capacity (aaa).  Identify the additional
capacity (aab).  Calculate the effective space taken by the stackable
crates (aae) - consider them as a single unit after stacking.
Calculate the new total capacity by adding the initial and additional
capacities (aac).  Finally, adjust the total capacity by considering
the space taken by the stackable crates (aaf). Question:

What is the new total storage capacity of the warehouse after these
adjustments?

(Use the given values and operations to solve the problem)
```

While the above application problem looks more vivid than the plain one, it is in fact challenging to verify its rationale and correctness. For instance, is the problem natural? Is the context correct? Note that GPT-4 tends to first compute the answers to the original problem before generating a new one, which may accumulate errors including problem understanding, application generation, and computation. Therefore, while we point the feasibility of such practice, this in current stage is difficult to verify and should be left for future work.

## C    PROOF

**Theorem C.1.** Given a tree-based DAG with depth $d$ and width $w$, if the operation set for non-leaf nodes has $k$ distinct operations and the value set for leaf nodes contains $n$ distinct values, the probability that two independently generated DAGs are identical is: $P = \left( k^{\frac{w^{d-1}-1}{w-1}} \times n^{w^{d-1}} \right)^{-1}$.

*Proof.* To determine the overall probability, we analyze the likelihood at each depth and then multiply these probabilities. For depth $i$, the number of nodes is $w^{i-1}$.

For depth $i, 1 \leq i \leq d-1$. Since these nodes are non-leaf nodes, the probability they are identical in two independently generated DAGs is the likelihood all of them have the same operations: $p_i = \frac{1}{k^{w^{i-1}}}$.

For the leaf nodes at depth $d$, the probability that they are the same across two DAGs is: $p_d = \frac{1}{n^{w^{d-1}}}$.

Thus, the overall probability $P$ that two DAGs are identical is: $P = \prod_{i=1}^{d-1} p_i \times p_d$.

Substituting the above expressions and simplifying gives the result: $P = \left( k^{\frac{w^{d-1}-1}{w-1}} \times n^{w^{d-1}} \right)^{-1}$.

Note: We consider two trees to be distinct even if they only differ in the order of operations. For instance, the tree representing $3 \times 5$ is considered different from the tree representing $5 \times 3$. Excluding such cases may be non-trivial and is unlikely to significantly affect the odds.

$\square$

**Theorem C.2.** Given a general DAG with $n$ nodes where each node has a minimum of $l \geq 1$ links, the lower bound of the probability that two randomly selected DAGs of this configuration are identical is $((n-1)!)^{-1}$.

*Proof.* Consider a DAG where each node has exactly one outgoing link. The first node can be connected to any of the remaining $n-1$ nodes. Subsequently, the second node can connect to any of the remaining $n-2$ nodes, excluding the one already connected to the first node. Following this logic, the third node can connect to any of the $n-3$ unconnected nodes, and so on.

Thus, the total number of distinct DAGs that can be constructed under these constraints is given by: $(n-1) \times (n-2) \times \ldots \times 2 \times 1 = (n-1)!$

Given two randomly chosen DAGs of this kind, the likelihood they are identical is the inverse of the number of unique DAGs: $\frac{1}{(n-1)!}$

This probability serves as the lower bound when considering the general case of DAGs with nodes having a minimum of $l \geq 1$ links, hence proving the theorem.

$\square$

## D    DETAILS OF EXPERIMENTS

### D.1    EXPERIMENT ENVIRONMENT

All experiments are conducted on a workstation equipped with an NVIDIA V100 GPU with 16GB memory and A100 GPU with 80GB memory. For GPT-3.5-Turbo and GPT-4, we use OpenAI's

API for inference, the versions are gpt-3.5-turbo-0613 and gpt-4-0613. For the Llama2 models, we downloaded from the Llama2 github repository[3] and follow the instruction[4] to convert them into huggingface models. For Vicuna-13B-v1.3, we downloaded it from its github repository[5]. The remaining models can be downloaded directly via huggingface.

## D.2    PROMPTS

- **Arithmetic:**

```
Here is a description of an arithmetic problem:
{}
Compute the result of {}.  If the solution cannot be calculated,
answer 'N/A'. Ensure your result is within a relative precision of
0.0001 (or 0.01%) compared to the ground truth value.  Ensure your
final result begins with '<<<' and ends with '>>>', for example, if
the answer is 1, your final result should be <<<1>>>.
```

- **Linear Equation:**

```
Given the following linear equation system with two variables:
{}
Determine the values of x and y.  Ensure your results are within a
relative precision of 0.001 (or 0.1%) compared to the ground truth
values.  Your response should be formatted as:  <<<x's value y's
value>>>, e.g., if x=1 and y=2, then it should be <<<1 2>>>
```

- **Boolean Logic:**

```
Here is a description of a boolean logic problem:
{}
Compute the result of {}.  If the solution can not be calculated,
answer 'N/A'. Ensure your final result begins with '<<<' and ends
with '>>>', for example, if the answer is True, your final result
should be <<<True>>>.
```

- **Deductive Logic:**

```
Here is a description of a deductive logic problem:
{}
The symbol '->' represents a deductive relationship, e.g., A -> B
implies that if A is true, then B is true.  If A is false, B's truth
value remains undetermined (N/A). Deduce the result of {}.  If the
solution can not be deduced, answer 'N/A'. Ensure your final result
begins with '<<<' and ends with '>>>', for example, if the answer is
True, your final result should be <<<True>>>.
```

- **Abductive Logic:**

```
Here is a description of an abductive logic problem:
{}
The symbol '->' represents a deductive relationship, e.g., A -> B
implies that if B is false, then A is false.  If B is true, A's
```

```
truth value remains undetermined (N/A). If the solution can not be
abduced, answer 'N/A'. Ensure your final result begins with '<<<'
and ends with '>>>', for example, if the answer is True, your final
result should be <<<True>>>.
```

- **Reachability:**

```
Given a directed graph:
{}
Respond with either '<<<True>>>' if reachable, or '<<<False>>>'
otherwise.
```

- **Max Sum Path:**

```
Given a directed graph with values assigned to each node:
{}
For exmaple, the value of the path A->B->C is obtained by summing
the values of nodes A, B, and C. Please format your response
as <<<Answer>>>.  For example, if the answer is 1, it should be
presented as <<<1>>>.
```

## D.3    EVALUATION SET

We categorize tasks into four complexity levels, denoted as D1 to D4. For tasks reliant on general Directed Acyclic Graphs (DAGs), the node count is set to $7, 10, 15, 20$. Each of these nodes possesses a maximum link range of $3, 4, 6, 8$ and a minimum link count of 1. Conversely, for tasks that utilize tree-based DAGs, the tree depths and widths are defined as $(2, 2), (3, 2), (3, 3), (4, 2)$, in order of increasing complexity.

The range of these datasets progresses from simple to intricate. To illustrate, an arithmetic problem with a tree depth of 2 represents a basic two-variable arithmetic computation. In contrast, a task with a tree depth of 4 exhibits heightened complexity, necessitating multiple inferential steps for resolution.

## D.4    DETAILS OF EXPERIMENT RESULTS

We do not report the results of Flan-T5-large, phi-1.5, WizardMath-13B, and Xwin-13B since their performance is almost 0 even on simplest evaluation sets generated by our DYVAL. Therefore, we extensively run the results of four remaining models: Vicuna-13B-v1.3, Llama2-13B-chat, GPT-3.5-Turbo, and GPT-4. Table 2, 3, and 4 report the detailed results (average±standard error) of these models in different complexity (D1~D4) and different description generation orders (topological, reversed topological, and random orders).

In the reachability task, as task difficulty escalated, Llama2-13B-chat's performance paradoxically improved. Upon investigation, Llama2-13B-chat essentially resorted to random guessing across datasets. The proportion of 'True' answers increased (from  40% in D1 to  60% in D3), with 'False' responses being nearly absent. The remainder were non-responses, thus elevating the overall accuracy. The observation is similar to those made in Sec.4.4 where we investigated the influence of different model size.

Further, generation description order affects outcomes: in the reachability task, GPT-4's accuracy drops by 13.67% when given reversed order compared to topological order. See Appendix D.4 for details of experiment results.

## D.5    DETAILS OF CASE STUDY

We select 20 failure cases of the most challenging datasets of arithmetic, deductive logic, abductive logic, and reachability of GPT-4.

Table 2: Results for Mathematic Tasks

| Task | Dataset | GPT4 | | | ChatGPT | | | Llama2-13b-chat | | | Vicuna-13b-v1.3 | | |
|---|---|---|---|---|---|---|---|---|---|---|---|---|---|
| | | Topo | Reversed | Rand | Topo | Reversed | Rand | Topo | Reversed | Rand | Topo | Reversed | Rand |
| Arithmetic | D1 | $98.00_{\pm0.00}$ | $100.00_{\pm0.00}$ | $99.00_{\pm1.00}$ | $95.00_{\pm0.40}$ | $99.53_{\pm0.23}$ | $97.27_{\pm0.90}$ | $12.33_{\pm0.90}$ | $38.67_{\pm1.86}$ | $24.20_{\pm1.93}$ | $2.73_{\pm0.64}$ | $0.53_{\pm0.58}$ | $2.40_{\pm0.69}$ |
| | D2 | $94.17_{\pm1.15}$ | $95.67_{\pm1.04}$ | $95.50_{\pm1.00}$ | $90.47_{\pm1.17}$ | $92.27_{\pm0.12}$ | $92.07_{\pm0.31}$ | $5.73_{\pm1.01}$ | $3.00_{\pm0.87}$ | $4.60_{\pm0.35}$ | $1.53_{\pm0.46}$ | $0.07_{\pm0.12}$ | $0.60_{\pm0.20}$ |
| | D3 | $85.83_{\pm1.89}$ | $87.67_{\pm1.61}$ | $84.35_{\pm2.26}$ | $76.20_{\pm2.80}$ | $78.20_{\pm3.41}$ | $78.47_{\pm3.83}$ | $1.07_{\pm0.12}$ | $2.47_{\pm0.23}$ | $3.07_{\pm0.76}$ | $1.13_{\pm0.31}$ | $0.07_{\pm0.12}$ | $0.20_{\pm0.00}$ |
| | D4 | $79.33_{\pm1.61}$ | $81.33_{\pm1.89}$ | $77.67_{\pm2.57}$ | $72.40_{\pm1.51}$ | $72.73_{\pm1.68}$ | $69.40_{\pm2.25}$ | $2.80_{\pm0.53}$ | $0.80_{\pm0.35}$ | $1.20_{\pm0.69}$ | $0.20_{\pm0.20}$ | $0.07_{\pm0.12}$ | $0.00_{\pm0.00}$ |
| Linear Equation | D1 | $56.33_{\pm1.15}$ | $58.50_{\pm0.00}$ | $56.33_{\pm3.01}$ | $36.20_{\pm1.04}$ | $36.20_{\pm2.42}$ | $36.27_{\pm2.66}$ | $0.00_{\pm0.00}$ | $0.00_{\pm0.00}$ | $0.00_{\pm0.00}$ | $0.00_{\pm0.00}$ | $0.00_{\pm0.00}$ | $0.00_{\pm0.00}$ |
| | D2 | $42.67_{\pm2.36}$ | $42.17_{\pm1.89}$ | $43.00_{\pm2.65}$ | $27.67_{\pm1.75}$ | $30.87_{\pm1.72}$ | $29.60_{\pm2.55}$ | $0.00_{\pm0.00}$ | $0.00_{\pm0.00}$ | $0.00_{\pm0.00}$ | $0.00_{\pm0.00}$ | $0.00_{\pm0.00}$ | $0.00_{\pm0.00}$ |
| | D3 | $44.33_{\pm2.52}$ | $43.17_{\pm6.60}$ | $43.83_{\pm2.93}$ | $19.40_{\pm1.06}$ | $29.67_{\pm1.29}$ | $23.87_{\pm2.10}$ | $0.00_{\pm0.00}$ | $0.00_{\pm0.00}$ | $0.00_{\pm0.00}$ | $0.00_{\pm0.00}$ | $0.00_{\pm0.00}$ | $0.00_{\pm0.00}$ |
| | D4 | $38.83_{\pm4.25}$ | $37.17_{\pm3.82}$ | $34.00_{\pm1.73}$ | $13.80_{\pm1.06}$ | $21.07_{\pm0.50}$ | $14.93_{\pm2.05}$ | $0.00_{\pm0.00}$ | $0.00_{\pm0.00}$ | $0.00_{\pm0.00}$ | $0.00_{\pm0.00}$ | $0.00_{\pm0.00}$ | $0.00_{\pm0.00}$ |

Table 3: Results for Logical Reasoning Tasks

| Task | Dataset | GPT4 | | | ChatGPT | | | Llama2-13b-chat | | | Vicuna-13b-v1.3 | | |
|---|---|---|---|---|---|---|---|---|---|---|---|---|---|
| | | Topo | Reversed | Rand | Topo | Reversed | Rand | Topo | Reversed | Rand | Topo | Reversed | Rand |
| Boolean Logic | D1 | $100.00_{\pm0.00}$ | $100.00_{\pm0.00}$ | $100.00_{\pm0.00}$ | $99.80_{\pm0.20}$ | $99.87_{\pm0.23}$ | $97.60_{\pm0.53}$ | $25.33_{\pm1.17}$ | $12.73_{\pm0.23}$ | $19.53_{\pm1.33}$ | $77.93_{\pm1.47}$ | $84.93_{\pm0.12}$ | $81.13_{\pm0.12}$ |
| | D2 | $100.00_{\pm0.00}$ | $99.33_{\pm0.58}$ | $100.00_{\pm0.00}$ | $98.80_{\pm0.20}$ | $99.40_{\pm0.20}$ | $96.80_{\pm0.40}$ | $7.87_{\pm0.61}$ | $17.00_{\pm1.40}$ | $18.67_{\pm0.70}$ | $43.00_{\pm1.00}$ | $68.40_{\pm2.12}$ | $53.93_{\pm1.55}$ |
| | D3 | $97.00_{\pm1.00}$ | $100.00_{\pm0.00}$ | $100.00_{\pm0.00}$ | $99.60_{\pm0.35}$ | $98.00_{\pm0.69}$ | $92.93_{\pm1.10}$ | $13.53_{\pm1.55}$ | $20.07_{\pm1.51}$ | $17.93_{\pm0.50}$ | $28.93_{\pm2.19}$ | $42.47_{\pm2.10}$ | $39.67_{\pm2.58}$ |
| | D4 | $96.00_{\pm2.00}$ | $100.00_{\pm0.00}$ | $99.67_{\pm0.58}$ | $99.40_{\pm0.20}$ | $95.47_{\pm0.23}$ | $90.47_{\pm0.64}$ | $10.87_{\pm0.42}$ | $13.93_{\pm1.30}$ | $16.33_{\pm0.99}$ | $29.20_{\pm1.59}$ | $29.73_{\pm2.12}$ | $29.80_{\pm1.25}$ |
| Deductive Logic | D1 | $100.00_{\pm0.00}$ | $88.17_{\pm1.26}$ | $95.17_{\pm1.53}$ | $81.87_{\pm0.76}$ | $82.47_{\pm1.42}$ | $81.53_{\pm2.72}$ | $45.40_{\pm1.25}$ | $56.27_{\pm1.03}$ | $49.13_{\pm0.42}$ | $11.87_{\pm0.31}$ | $44.60_{\pm1.11}$ | $20.73_{\pm0.99}$ |
| | D2 | $98.50_{\pm1.50}$ | $92.50_{\pm0.87}$ | $97.17_{\pm1.61}$ | $64.60_{\pm1.60}$ | $65.93_{\pm1.14}$ | $63.73_{\pm3.42}$ | $34.60_{\pm2.60}$ | $34.47_{\pm2.05}$ | $43.07_{\pm3.14}$ | $48.00_{\pm2.31}$ | $38.73_{\pm2.05}$ | $44.87_{\pm0.61}$ |
| | D3 | $98.17_{\pm1.53}$ | $87.83_{\pm2.52}$ | $98.33_{\pm1.04}$ | $63.47_{\pm2.48}$ | $61.60_{\pm2.80}$ | $63.33_{\pm1.86}$ | $26.60_{\pm1.91}$ | $33.47_{\pm1.68}$ | $26.27_{\pm1.21}$ | $46.67_{\pm1.75}$ | $45.47_{\pm2.72}$ | $34.67_{\pm2.21}$ |
| | D4 | $96.17_{\pm1.04}$ | $84.33_{\pm1.44}$ | $90.67_{\pm5.03}$ | $56.40_{\pm1.78}$ | $57.33_{\pm1.30}$ | $56.47_{\pm3.00}$ | $20.60_{\pm1.56}$ | $29.20_{\pm1.59}$ | $20.60_{\pm2.69}$ | $38.07_{\pm1.15}$ | $37.40_{\pm1.22}$ | $33.40_{\pm3.17}$ |
| Abductive Logic | D1 | $93.50_{\pm0.50}$ | $83.33_{\pm3.33}$ | $91.00_{\pm1.00}$ | $37.93_{\pm2.14}$ | $49.33_{\pm3.59}$ | $38.07_{\pm2.61}$ | $3.73_{\pm0.23}$ | $0.00_{\pm0.00}$ | $1.73_{\pm0.76}$ | $56.40_{\pm2.25}$ | $31.60_{\pm1.22}$ | $45.53_{\pm2.10}$ |
| | D2 | $78.83_{\pm6.37}$ | $48.50_{\pm5.57}$ | $63.50_{\pm4.09}$ | $53.47_{\pm2.50}$ | $59.80_{\pm3.41}$ | $56.60_{\pm3.36}$ | $21.47_{\pm1.17}$ | $10.53_{\pm1.42}$ | $17.67_{\pm1.86}$ | $19.80_{\pm0.20}$ | $25.47_{\pm1.72}$ | $22.00_{\pm1.39}$ |
| | D3 | $64.67_{\pm5.51}$ | $49.83_{\pm3.18}$ | $58.50_{\pm3.28}$ | $56.13_{\pm3.06}$ | $60.80_{\pm1.06}$ | $57.87_{\pm2.81}$ | $12.60_{\pm1.51}$ | $7.60_{\pm2.25}$ | $8.07_{\pm0.95}$ | $20.40_{\pm1.31}$ | $14.80_{\pm0.92}$ | $17.20_{\pm0.87}$ |

Here we present one failure case for each error type.

**Partial calculation error**   It has been observed that GPT-4, in certain situations, commits errors in intermediate computational steps, while it maintains correctness in the remaining steps. This characteristic anomaly is not isolated to complex calculations. In fact, not only complex calculation, seemingly straightforward calculations such as can be incorrectly computed. This observed behavior is consistent with the findings presented by (Dziri et al., 2023), where they highlighted that low-level learning models (LLMs) occasionally produce results that are only partially accurate, particularly in the realm of multi-digit multiplication.

```
Input:
Here is a description of an arithmetic problem:

The value of aaj is 7.

aak gets its value by squaring the value that aaj has.

The value of aah is 6.

The value of aag is 2.

aai gets its value by dividing the value of aag by those of aah.

aan gets its value by multiplying together the value of aai and aak.

The value of aaa is 6.

aab gets its value by squaring the value that aaa has.

The value of aac is 8.

The value of aad is 1.
```

Table 4: Results for Algorithm Tasks

| Task | Dataset | GPT4 | | | ChatGPT | | | Llama2-13b-chat | | | Vicuna-13b-v1.3 | | |
|---|---|---|---|---|---|---|---|---|---|---|---|---|---|
| | | Topo | Reversed | Rand | Topo | Reversed | Rand | Topo | Reversed | Rand | Topo | Reversed | Rand |
| Reachability | D1 | $83.67_{\pm1.15}$ | $92.67_{\pm1.15}$ | $85.33_{\pm3.06}$ | $59.53_{\pm0.76}$ | $63.87_{\pm1.51}$ | $63.40_{\pm2.42}$ | $21.60_{\pm1.20}$ | $23.20_{\pm0.80}$ | $26.87_{\pm1.47}$ | $11.47_{\pm1.29}$ | $29.80_{\pm2.03}$ | $23.53_{\pm2.20}$ |
| | D2 | $85.00_{\pm0.00}$ | $91.00_{\pm3.00}$ | $83.00_{\pm2.00}$ | $53.53_{\pm3.97}$ | $56.73_{\pm2.81}$ | $54.27_{\pm1.79}$ | $34.60_{\pm1.40}$ | $26.87_{\pm1.27}$ | $26.27_{\pm1.17}$ | $12.07_{\pm0.58}$ | $31.73_{\pm0.64}$ | $21.73_{\pm0.50}$ |
| | D3 | $68.17_{\pm2.93}$ | $77.67_{\pm0.58}$ | $67.67_{\pm2.31}$ | $49.67_{\pm2.55}$ | $57.53_{\pm1.90}$ | $53.73_{\pm3.13}$ | $39.33_{\pm2.08}$ | $39.33_{\pm1.80}$ | $37.47_{\pm1.01}$ | $13.60_{\pm1.00}$ | $29.67_{\pm3.75}$ | $21.80_{\pm2.88}$ |
| | D4 | $63.00_{\pm1.00}$ | $76.67_{\pm0.58}$ | $74.33_{\pm2.52}$ | $49.40_{\pm2.71}$ | $59.13_{\pm3.49}$ | $52.33_{\pm2.89}$ | $33.67_{\pm2.97}$ | $41.53_{\pm1.22}$ | $38.60_{\pm0.92}$ | $10.80_{\pm0.72}$ | $30.73_{\pm1.53}$ | $22.00_{\pm1.64}$ |
| Max Sum Path | D1 | $37.33_{\pm5.86}$ | $30.67_{\pm6.11}$ | $26.67_{\pm6.03}$ | $29.47_{\pm2.08}$ | $29.67_{\pm2.77}$ | $25.27_{\pm0.83}$ | $0.00_{\pm0.00}$ | $0.00_{\pm0.00}$ | $0.00_{\pm0.00}$ | $0.00_{\pm0.00}$ | $0.00_{\pm0.00}$ | $0.00_{\pm0.00}$ |
| | D2 | $38.67_{\pm8.14}$ | $27.00_{\pm8.54}$ | $25.67_{\pm5.69}$ | $14.20_{\pm1.25}$ | $13.33_{\pm2.37}$ | $11.33_{\pm0.76}$ | $0.00_{\pm0.00}$ | $0.00_{\pm0.00}$ | $0.00_{\pm0.00}$ | $0.00_{\pm0.00}$ | $0.00_{\pm0.00}$ | $0.00_{\pm0.00}$ |
| | D3 | $21.33_{\pm4.04}$ | $17.00_{\pm5.00}$ | $16.67_{\pm4.16}$ | $6.40_{\pm1.64}$ | $8.40_{\pm1.51}$ | $7.60_{\pm0.72}$ | $0.00_{\pm0.00}$ | $0.00_{\pm0.00}$ | $0.00_{\pm0.00}$ | $0.00_{\pm0.00}$ | $0.00_{\pm0.00}$ | $0.00_{\pm0.00}$ |
| | D4 | $16.67_{\pm2.52}$ | $12.00_{\pm2.00}$ | $10.67_{\pm0.58}$ | $5.07_{\pm0.76}$ | $6.20_{\pm1.11}$ | $6.67_{\pm1.33}$ | $0.00_{\pm0.00}$ | $0.00_{\pm0.00}$ | $0.00_{\pm0.00}$ | $0.00_{\pm0.00}$ | $0.00_{\pm0.00}$ | $0.00_{\pm0.00}$ |

aae gets its value by dividing the value of aac by those of aad.

aaf gets its value by dividing the value of aab by those of aae.

aao gets its value by adding together the value of aaf and aan.

Compute the result of aao.  If the solution cannot be calculated, answer 'N/A'. Ensure your result is within a relative precision of 0.0001 (or 0.01%) compared to the ground truth value.  Ensure your final result begins with '<<<' and ends with '>>>', for example, if the answer is 1, your final result should be <<<1>>>.

**GPT-4's response:**
First, let's calculate the values step by step:

aaj = 7

aak = aaj$^2$ = 7$^2$ = 49

aah = 6

aag = 2

aai = aag / aah = 2 / 6 = 0.3333

aan = aai * aak = 0.3333 * 49 = 16.3167 (should be 16.3317)

aaa = 6

aab = aaa$^2$ = 6$^2$ = 36

aac = 8

aad = 1

aae = aac / aad = 8 / 1 = 8

aaf = aab / aae = 36 / 8 = 4.5

aao = aaf + aan = 4.5 + 16.3167 = 20.8167

So, the result of aao is 20.8167.

Therefore, the final result is <<<20.8167>>>.

**Ground truth:** 20.8333

**Incorrect reasoning**    When tasked with reasoning, GPT-4 can occasionally falter in rule interpretation, especially in abductive logic task.

**Input:**
Here is a description of an abductive logic problem:

(aaa and aab) -> aac.

(aae or aaf) -> aag.

(aah or aai) -> aaj.

(aag or aaj) -> aak.

(NOT aac) -> aad.

(aad or aak) -> aan.

Given aan is False, what is the value of aab?

The symbol '->' represents a deductive relationship, e.g., A -> B implies that if B is false, then A is false.  If B is true, A's truth value remains undetermined (N/A). If the solution can not be abduced, answer 'N/A'. Ensure your final result begins with '<<<' and ends with

'>>>', for example, if the answer is True, your final result should be
<<<True>>>.

**GPT-4's response:**
Given aan is False, it means either aad or aak must be False (both aad
and aak should be False) (since aad or aak -> aan).

If aak is False, then either aag or aaj must be False (since aag or
aaj -> aak).

If aag is False, then either aae or aaf must be False (since aae or
aaf -> aag).

If aaj is False, then either aah or aai must be False (since aah or
aai -> aaj).

However, none of these conditions provide any information about aab.

Therefore, the value of aab cannot be abduced from the given
information.

So, the answer is <<<N/A>>>.

**Ground truth:** N/A

**Self Contradiction** GPT-4, on occasion, presents self-contradictory assumptions during the same inference in an abductive logic task.

**Input:**
Here is a description of an abductive logic problem:

(aah or aai) -> aaj.

(NOT aaj) -> aak.

(aad or aae) -> aaf.

(aaa or aab) -> aac.

(aac or aaf) -> aag.

(aag or aak) -> aan.

Given aan is False, what is the value of aah?

The symbol '->' represents a deductive relationship, e.g., A -> B
implies that if B is false, then A is false.  If B is true, A's truth
value remains undetermined (N/A). If the solution can not be abduced,
answer 'N/A'. Ensure your final result begins with '<<<' and ends with
'>>>', for example, if the answer is True, your final result should be
<<<True>>>.

**GPT-4's response:**
Since aan is False, at least one of aag or aak must be False.
However, we don't know which one is False, so we can't determine the
value of aah.  (This inference step contradicts to the former example,
where GPT-4 assumes aah is False.)  Therefore, the answer is 'N/A'.

<<<N/A>>>

**Ground truth:** N/A

**Unsubstaintiated response** A recurring observation in both reasoning and algorithmic tasks is GPT-4's propensity to provide answers without accompanying justifications. These succinct, answer-only responses raise concerns about the depth of GPT-4's understanding and whether its approach is grounded in actual comprehension or simply a result of memorization. This error mostly happens in algorithm tasks and occasionally happens in logical reasoning tasks.

**Input:**
```
Here is a description of an abductive logic problem:

(aaa or aab) -> aac.

(aad or aae) -> aaf.

(aac or aaf) -> aag.

(NOT aah) -> aai.

(NOT aaj) -> aak.

(aai or aak) -> aan.

(aag or aan) -> aao.

Given aao is False, what is the value of aaa?

The symbol '->' represents a deductive relationship, e.g., A -> B
implies that if B is false, then A is false.  If B is true, A's truth
value remains undetermined (N/A). If the solution can not be abduced,
answer 'N/A'. Ensure your final result begins with '<<<' and ends with
'>>>', for example, if the answer is True, your final result should be
<<<True>>>.
```

**GPT-4's response:**
<<<N/A>>>

**Ground truth:** N/A

**Instructional oversight** Interestingly, while GPT-4 often computes accurately, it sometimes falters when it comes to precisely following the output instructions as described in given prompts. This discrepancy points to a potential gap in the model's ability to parse and adhere to intricate instructional details, even when the underlying computational capability is sound.

**Input:**
```
Here is a description of an arithmetic problem:

The value of aac is 6.

The value of aai is 6.

The value of aag is 8.

aan gets its value by subtracting the sum of the values of aak and aae
from the value of aah.

aao gets its value by multiplying together the value of aae and aan
and aae.

aah gets its value by multiplying together the value of aaf and aag
and aae.

The value of aap is 1.

The value of aaa is 6.

aad gets its value by taking the square root of the value that aac
has.

The value of aaj is 4.

aak gets its value by subtracting the sum of the values of aaj and aae
from the value of aai.

The value of aaq is 7.

aar gets its value by dividing the value of aap by those of aaq.
```

```
aab gets its value by squaring the value that aaa has.

The value of aaf is 10.

aae gets its value by multiplying together the value of aab and aad
and aai.

Compute the result of aao.  If the solution cannot be calculated,
answer 'N/A'. Ensure your result is within a relative precision of
0.0001 (or 0.01%) compared to the ground truth value.  Ensure your
final result begins with '<<<' and ends with '>>>', for example, if
the answer is 1, your final result should be <<<1>>>.
```

**GPT-4's response:**
```
Let's solve this step by step:

1.  aac = 6

2.  aai = 6

3.  aag = 8

4.  aaa = 6

5.  aad = sqrt(aac) = sqrt(6)

6.  aaj = 4

7.  aaq = 7

8.  aap = 1

9.  aaf = 10
```
10.  aab = aaa$^2$ = 6$^2$ = 36
```
11.  aae = aab * aad * aai = 36 * sqrt(6) * 6

12.  aak = aai - (aaj + aae) = 6 - (4 + 36 * sqrt(6) * 6)

13.  aah = aaf * aag * aae = 10 * 8 * 36 * sqrt(6) * 6

14.  aan = aah - (aak + aae) = 10 * 8 * 36 * sqrt(6) * 6 - (6 - (4 +
36 * sqrt(6) * 6) + 36 * sqrt(6) * 6)

15.  aao = aae * aan * aae = (36 * sqrt(6) * 6) * (10 * 8 * 36 *
sqrt(6) * 6 - (6 - (4 + 36 * sqrt(6) * 6) + 36 * sqrt(6) * 6)) * (36 *
sqrt(6) * 6)
```
```
This is a complex calculation and it's not possible to simplify it
further.  However, it's clear that the result will be a very large
number.  Since the problem doesn't provide any constraints on the
values of the variables, we can't say that the solution is 'N/A'.
(The equation is correct however GPT-4 did not simplify it.  In our
experiment, we found that GPT-4 can handle large numbers, so the error
does not originate from them.)

So, the result of aao is a very large number that can't be simplified
further.
```

**Ground truth:** 11848342359.78579

## D.6 DETAILS OF VARING COMPLEXITY CONSTRAINTS

As shown in Figure 5, we systematically vary the levels of complexity in GPT-3.5-Turbo by adjusting individual constraints while keeping others constant. Specifically, we explore how performance metrics evolve as we incrementally adjust depth, width, #nodes, #max links, the number of extra links, and the quantity of random descriptions across arithmetic, boolean logic, and deductive logic tasks. To comprehensively evaluate the impact of complexity constraints, various parameters were meticulously adjusted. The following elucidates the configurations employed:

Table 5: Results of GPT-3.5-Turbo with prompt engineering techniques on the toughest evaluation sets generated by DYVAL (D4).

| Prompt engineering | Arithmetic | Linear Equation | Deductive Logic | Abductive Logic | Reachability | Max Sum Path |
|---|---|---|---|---|---|---|
| Vanilla | 42.13 | 14.93 | 56.40 | 54.33 | 49.40 | 5.07 |
| CoT (Wei et al., 2022) | 42.33 | 21.93 | 52.93 | 43.73 | 47.73 | 1.93 |
| Fewshot (Brown et al., 2020) | 47.86 | 2.40 | 35.93 | 41.60 | 81.80 | 12.20 |
| Least2most (Zhou et al., 2023b) | 36.73 | 12.47 | 44.07 | 38.80 | 76.53 | 8.07 |
| APE (Zhou et al., 2023d) | 45.20 | 23.40 | 44.67 | 53.13 | 62.80 | 8.87 |
| SKiC (Chen et al., 2023) | 32.07 | 13.70 | 63.00 | 78.27 | 71.40 | 11.80 |

Table 6: Results of Llama 2 with different sizes on DY-VAL-generated evaluation samples (D1).

| Size | Arithmetic | Boolean logic | Reachability |
|---|---|---|---|
| 7b | 13.07 | 28.93 | 29.53 |
| 13b | 24.20 | 19.53 | 26.53 |
| 70b | 29.71 | 28.30 | 47.38 |

- **Depth Constraint:** Maintaining the width at 2, with neither the addition of random links nor the embedding of extra descriptions (both set to 0), the depth was systematically varied, with values set to $2, 3, 4, 5$, and $6$.

- **Width Constraint:** With a fixed depth of 3, and with the addition of random links and embedding of extra descriptions both neutralized to 0, the width was tested with the values $2, 3, 4, 5$, and $6$.

- **Random Link Addition Constraint:** For this, a depth of 4 and a width of 2 were maintained, with extra descriptions set to 0. The number of random links introduced varied as $0, 1, 2, 3$, and $4$. It should be highlighted that due to the inherent acyclic constraint, certain nodes may preclude the addition of extra links.

- **Embedding Extra Descriptions:** With a depth and width fixed at 4 and 2, respectively, and no addition of random links (set to 0), the levels of embedded extra descriptions were calibrated to $0, 1, 2, 3$, and $4$.

Across these variations, our results consistently underscore a notable trend: as the tasks become more intricate through the augmentation of these complexity parameters, LLMs progressively struggle, underscoring the inherent challenges posed by increasing task intricacy. It can be observed that depth is the most influential complexity constraint of tree-based DAGs, indicates that LLMs struggle to deal with problems that requires more inference steps.

## D.7 DETAILS OF PROMPT ENGINEERING

We explored five prompting techniques to evaluate their potential impact on our most challenging datasets (excluding boolean logic since GPT-3.5-Turbo achieved comparable results on most challenging datasets): Zeroshot-CoT (Wei et al., 2022), Few-shot (3-shot in our experiments) (Brown et al., 2020), Least-to-most (Zhou et al., 2023b), automatic prompt engineering (APE) (Zhou et al., 2023d), and skill-in-context (SkiC) (Chen et al., 2023). The details of these techniques are as follows:

- **Zeroshot-CoT:** An approach that allows models to generalize from their pre-training without explicit examples in the target task (Wei et al., 2022).

- **Fewshot (3-shot in our experiments):** Provides the model with a small number of examples from the target task to aid in understanding and generalizing to the broader task (Brown et al., 2020).

- **Least to Most Prompting:** This technique incrementally provides more specific prompts to guide the model's responses, adapting the prompt based on the difficulty level of the problem (Zhou et al., 2023b).

- **Automatic Prompting Engineering (APE):** A method where prompts are automatically engineered to elicit the desired response from the model, often maximizing its performance (Zhou et al., 2023d).

- **Skill-in-Context (SKiC):** This method seeks to understand a model's inherent skills and utilize them in a specific context to improve its outputs (Chen et al., 2023).

## D.8 HUMAN STUDY

We conducted our human study by obeying the local laws and regulations. The demographics of the human subjects are shown in Table 7.

Table 7: Demographics of the recruited human subjects.

| Sex | Age | Degree |
|---|---|---|
| Male: 63 (63%) | 20-25: 75 (75%) | Bachelor: 62 (62%) |
| Female: 37 (37%) | 26-36: 25 (25%) | Master: 38 (38%) |

# E  DETAILS OF FINE-TUNING

## E.1  CONSTRUCTING TRAINING DATA

The training data includes the problems and their corresponding inference steps. Constructing inference steps is similar to the DAG description. For mathematics tasks and logical reasoning tasks, which utilize T-DAG, we follow the topological order to describe how the intermediate gets its value from its children. For algorithm tasks, the inference are indeed an algorithm, thus, we formalize each step in the algorithm into natural language, which finally form the inference steps. Following list presented a training sample of each task. Note that the inference steps for each task are also highly flexible and can be augmented with more details, and we believe that the more detailed inference steps will benefit more.

- **Arithmetic:**

```
Input:
Here is a description of an arithmetic problem:
The value of aae is 2.
The value of aad is 4.
aaf gets its value by dividing the value of aad by those of aae.
aag gets its value by taking the square root of the value that aaf
has.
Compute the result of aag.  If the solution cannot be calculated,
answer 'N/A'. Ensure your result is within a relative precision of
0.0001 (or 0.01%) compared to the ground truth value.  Ensure your
final result begins with '<<<' and ends with '>>>', for example, if
the answer is 1, your final result should be <<<1>>>.
Inference steps:
aae is 2.0
aad is 4.0
aaf = aad / aae = 4.0 / 2.0 = 2.0
aag = sqrt aaf = sqrt(2.0) = 1.41421356

Thus, the answer is <<<1.41421356>>>
```

- **Linear Equation:**

```
Input:
Given the following linear equation system with two variables:

aag0 x + -6 y = -6
-2 x + 6 y = 0

The calculation of aag0 is defined as:
The value of aab0 is 10.
The value of aaa0 is 6.
aac0 gets its value by adding together the value of aaa0 and aab0.
The value of aad0 is 6.
The value of aae0 is 1.
aaf0 gets its value by multiplying together the value of aad0 and
```

```
aae0.
aag0 gets its value by multiplying together the value of aac0 and
aaf0.

Determine the values of x and y.  Ensure your results are within
a relative precision of 0.001 (or 0.1%) compared to the ground truth
values.  Your response should be formatted as:  <<<x's value y's
value>>>, e.g., if x=1 and y=2, then it should be <<<1 2>>>
```
**Inference steps:**
```
Let's first solve the coefficients of the linear equation.
aab0 is 10.0
aaa0 is 6.0
aac0 = aaa0 + aab0 = 6.0 + 10.0 = 16.0
aad0 is 6.0
aae0 is 1.0
aaf0 = aad0 * aae0 = 6.0 * 1.0 = 6.0
aag0 = aac0 * aaf0 = 16.0 * 6.0 = 96.0
Thus, aag0 = 96.0

Next, solve the linear equation:

96.0 x + -6.0 y = -6.0
-2.0 x + 6.0 y = 0.0
To eliminate 'y', multiply the first equation by 6 and the
second equation by -6.  This makes the coefficients of 'y' equal.
Subtracting the second equation from the first then gives:  564.0x =
-36.0.
From the equation 564.0x = -36.0, we can solve for x.
Solving for x, we get x = -0.06382979.
Substituting x = -0.06382979 into the first original equation, we
get:  -6.0y = 0.12765957, which gives y = -0.0212766.

Thus, the answer is <<<-0.06382979 -0.0212766>>>
```

• **Boolean Logic:**

**Input:**
```
Here is a description of a boolean logic problem:
aaa is False.
aab is True.
The value of aac equals to (aaa OR aab).
aad is False.
aae is False.
The value of aaf equals to (aad AND aae).
The value of aag equals to (aac AND aaf).
Compute the result of aag.  If the solution can not be calculated,
answer 'N/A'. Ensure your final result begins with '<<<' and ends
with '>>>', for example, if the answer is True, your final result
should be <<<True>>>.
```
**Inference steps:**
```
aaa is False.
aab is True.
aac = (aaa OR aab) = (False OR True) = True.
aad is False.
aae is False.
aaf = (aad AND aae) = (False AND False) = False.
```

```
aag = (aac AND aaf) = (True AND False) = False.

Thus, the answer is <<<False>>>
```

- **Deductive Logic:**

**Input:**
Here is a description of a deductive logic problem:
aaa is False.
aab is False.
(aaa or aab) -> aac.
(NOT aac) -> aad.
The symbol '->' represents a deductive relationship, e.g., A -> B
implies that if A is true, then B is true.  If A is false, B's truth
value remains undetermined (N/A). Deduce the result of aad.  If the
solution can not be deduced, answer 'N/A'. Ensure your final result
begins with '<<<' and ends with '>>>', for example, if the answer is
True, your final result should be <<<True>>>.
**Inference steps:**
aaa is False.
aab is False.
(aaa OR aab) = (False OR False) -> aac.
The premise (False OR False) is False, thus, the value of aac cannot
be deduced and is set to N/A.
(NOT aac) = (NOT N/A) -> aad.
The value of aad cannot be deduced, and is set to N/A.

Thus, the answer is <<<N/A>>>

- **Abductive Logic:**

**Input:**
Here is a description of an abductive logic problem:
(aad or aae) -> aaf.
(aaa or aab) -> aac.
(aac or aaf) -> aag.
Given aag is False, what is the value of aab?
The symbol '->' represents a deductive relationship, e.g., A -> B
implies that if B is false, then A is false.  If B is true, A's
truth value remains undetermined (N/A). If the solution can not be
abduced, answer 'N/A'. Ensure your final result begins with '<<<'
and ends with '>>>', for example, if the answer is True, your final
result should be <<<True>>>.
**Inference steps:**
(aac OR aaf) -> aag = False.
Given aag is False, the value of premise (aac OR aaf) is False,
thus, the value of aac is abduced as False.
(aaa OR aab) -> aac = False.
Given aac is False, the value of premise (aaa OR aab) is False,
thus, the value of aab is abduced as False.

Thus, the answer is <<<False>>>

- **Reachability:**

**Input:**
```
Given a directed graph:
aag points to:  (None).
aaf points to:  (None).
aaa points to:  (aag, aaf).
aac points to:  (aaa).
aab points to:  (aaa).
aad points to:  (aaa).
aae points to:  (aad).
Can aag be reached starting from aaf?
Respond with either '<<<True>>>' if reachable, or '<<<False>>>'
otherwise.
```
**Inference steps:**
```
Starting the search process from node aaf with the goal to reach
node aag.
Checking node aaf.
Node aaf has no unvisited children.  Moving back.
Exhausted all possible paths without reaching node aag.

Thus, the answer is <<<False>>>
```

- **Max Sum Path:**

**Input:**
```
Given a directed graph with values assigned to each node:
aag points to:  (None).
aaf points to:  (aag).
aae points to:  (aaf, aag).
aab points to:  (aaf).
aad points to:  (aab, aae).
aac points to:  (aag, aad).
aaa points to:  (aac, aag, aae).
The value of aaa is 3
The value of aab is 2
The value of aaf is 5
The value of aag is 5
The value of aae is 1
The value of aad is 7
The value of aac is 2
What's the maximum sum path from aab to aae?
For exmaple, the value of the path A->B->C is obtained by summing
the values of nodes A, B, and C. Please format your response
as <<<Answer>>>.  For example, if the answer is 1, it should be
presented as <<<1>>>.
```
**Inference steps:**
```
Starting the search for the maximum sum path from node aab to node
aae.
Reaching node aab with current sum of 2.
Now, we explore the children of node aab:  aaf.
Reaching node aaf with current sum of 7.
Now, we explore the children of node aaf:  aag.
Reaching node aag with current sum of 12.
Node aag has no children.  Moving back.
There is no path from node aab to node aae.

Thus, the answer is <<<N/A>>>
```

### E.2 Training data and testing data

#### E.2.1 Training data

For mathematical tasks and logical reasoning tasks that utilize T-DAGs, we construct four types of training datasets. Each dataset consists of 500 training samples. All of these types have a depth of 3. The settings are as follows:

1. width=2, add random links=0, embed random descs=0,
2. width=2, add random links=1, embed random descs=0,
3. width=2, add random links=1, embed random descs=1,
4. width=3, add random links=0, embed random descs=0.

For algorithm tasks, two types of training datasets are generated:

1. num nodes=7, max links per node=3,
2. num nodes=10, max links per node=4.

#### E.2.2 Testing data

We create three types of testing data:

1. **In-Distribution (ID) Test Set:** The difficulty level matches that of the training set.
   - For T-DAGs: depth=4, width=2, with no extra links and random descriptions.
   - For G-DAGs: num nodes=15 with max links=6.
2. **Out-of-Distribution (OOD) Test Set:**
   - For T-DAGs: depth=4, width=2, without extra links and random descriptions.
   - For G-DAGs: num nodes=15 with max links=6.
3. **Out-of-Distribution-Hard (OOD-hard) Test Set:**
   - For T-DAGs: depth=4, width=2, with one extra link per node and one random description.
   - For G-DAGs: num nodes=20 with max links=8.

Note that the definition of OOD in our tasks is mainly on the different complexities of the samples that may come with more advanced structures or descriptions. For model evaluation, when using the DYVAL generated testing data, a zero-shot setting was adopted. For existing benchmarks, few-shot COT examples were provided in the context: 4 examples for GSM8K and SVAMP, 3 for FOLIO and RACO, and 2 for DP and LCS. The results of evaluation in our tasks are presented in Figure 10.

### E.3 Results of Fine-tuning

We fine-tuned Llama2-13b-chat with LORA (Hu et al., 2022) for 3 epochs where the rank was $8$, the scaling factor was $16$ and the drop out rate was $0.05$. We used a $0.0003$ learning rate with batch size $128$. Results on existing benchmarks of the fine-tuned model is in Figure 6 of the main paper.

Figure 10 displays the results after fine-tuning on our test datasets as described in Sec.E.2.2. The performance of Llama2-13B-chat on tasks like boolean logic, deductive logic, and reachability significantly improves after fine-tuning on our dataset. However, noticeable gaps remain, particularly in areas such as mathematic tasks, abductive logic and the max sum path.

## F Imbalanced Generated Dataset

Our algorithm can easily satisfy the balance requirement by meticulously controlling the flexible dynamic generation process. For example, in reachability task, we can drop the generated evaluation samples with 'False' labels until we generate a sample with 'True' label. We presented the results of GPT-3.5-Turbo and GPT-4 in balanced datasets in Table 8. The results in balanced datasets are similar to our initial findings: (1) GPT-3.5-Turbo consistently predicted all questions as "True.", resulted in a uniform accuracy rate of 50%. (2) GPT-4 demonstrated excellent performance. It maintained significantly higher accuracy rates across all complexity levels.

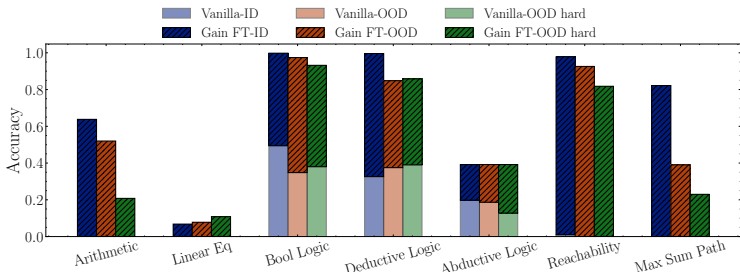

Figure 10: Fine-tuned results on ID and OOD sets of our tasks. For arithmetic, linear equation, reachability and max sum path tasks, the vanilla accuracy is zero.

Table 8: Results of fine-tuned ChatGPT model on general langugage understanding tasks

| Model | ChatGPT | | | | GPT4 | | | |
|---|---|---|---|---|---|---|---|---|
| Complexity | D1 | D2 | D3 | D4 | D1 | D2 | D3 | D4 |
| Balanced | 50 | 50 | 50 | 50 | 84.54 | 79.03 | 73.5 | 72.41 |
| Imbalanced | 63.87 | 54.27 | 53.73 | 52.33 | 85.33 | 83 | 67.67 | 74.33 |

# G GENERAL LANGUAGE UNDERSTANDING ABILITY AFTER FINE-TUNING

We fine-tuned GPT-3.5-turbo-0613 using our generated data on abductive logic and reachability datasets, as GPT-3.5 performs worst on these two datasets. Specifically, we generated 100 samples across complexity levels D1, D2, and D3 for each task. We compared the performance of original and fine-tuned models on several benchmark tasks in GLUE dataset. The performance of abductive logic and reachability task are tested on D4 task (different from fine-tuning dataset). As shown in Table 9, performance on WNLI and QNLI datasets dropped for the fine-tuned model. However, fine-tuned model achieves better results on CoLA, QQP, and MRPC datasets. Despite the mixed results, the overall improvement in several datasets suggests that fine-tuning on our generated datasets does not necessarily hurt the general language understanding ability.

Table 9: Results of fine-tuned ChatGPT model on general langugage understanding tasks

| | Abductive Logic | Reachability | SST-2 | CoLA | WNLI | QNLI | QQP | MRPC |
|---|---|---|---|---|---|---|---|---|
| GPT3.5 | 55.27 | 50.00 | 93.29 | 77.00 | 59.15 | 80.00 | 76.50 | 73.0 |
| GPT3.5-FT | 85.10 | 96.53 | 93.23 | 78.00 | 45.07 | 72.50 | 78.00 | 77.5 |

# H FLEXIBILITY TO NATURAL LANGUAGE TASKS

Finally, we discuss the flexibility of DYVAL while the main focus of this paper is on reasoning tasks. We show that DYVAL can be easily extended to natural language processing tasks using an initial experiment on sentiment analysis.

Generally speaking, a natural language sentence can be expressed as a syntax tree, similar to DAGs. However, generating sentences through direct syntax tree construction (which is similar to the construction of arithmetic task) presents notable challenges, primarily due to the need for grammatical correctness and the inherent naturalness of these sentences. Nevertheless, DYVAL can still be applied to generate tasks in natural language by utilizing syntax tree templates extracted by existing sentences. For each sentence in the SST-2 dataset, we initially employ GPT-3.5-Turbo to extract its syntactic structure. Within each syntax tree (i.e., DAGs), we identify the elements that can be modified: namely, nouns (such as names and places) and adjectives. GPT-3.5-Turbo is then used to create five alternative candidates for each of these modifiable components, which are subsequently replaced in an iterative fashion. Throughout this process, we continuously assess whether these replacements alter the original semantic meaning of the sentence. Any changes that result in a semantic shift are discarded. Note that the graph cannot be randomly generated as the reasoning tasks

since we need to constrain the naturalness and grammar correctness of the generated sentences. As a remedy, the structure of the graph can be abstracted using the template sentences generated by GPT-3.5-Turbo.

We generate three alternative versions for each sentence in the above process, forming our newly generated dataset. We then evaluate the performance of both Flan-T5-large and Llama2-7b models, using the original SST-2 dataset as well as our generated dataset for comparison. The results of these evaluations are detailed in Table 10. It shows that using our generated samples, the performance drops, indicating that we are creating challenging test sets. Note that this is an initial study and extending DYVAL to NLP tasks is nontrivial that cannot be covered in this paper, but should be left for future work.

Table 10: Results for Logical Reasoning Tasks

|        | Flan-T5-large | Llama2-7b |
|--------|---------------|-----------|
| Origin | **93.12**     | **90.37** |
| DyVal  | 86.46         | 72.03     |

