# OpenReview forum: "DyVal: Dynamic Evaluation of Large Language Models for Reasoning Tasks"
_ICLR.cc/2024/Conference — ICLR 2024 spotlight_

### Official Review · Reviewer_ydTt · 2023-10-19

**Soundness:** 3 good
**Presentation:** 3 good
**Contribution:** 3 good
**Rating:** 8
**Confidence:** 3

**Summary:**

The paper introduces DYVAL, a novel and flexible evaluation protocol for assessing Large Language Models (LLMs). DYVAL addresses two fundamental challenges in current LLM evaluation: potential data contamination in training data and the static nature of existing benchmarks that inadequately gauge LLMs' evolving capabilities. DYVAL dynamically generates evaluation samples using directed acyclic graphs (DAGs), allowing for controllable complexities in reasoning tasks. The authors evaluate various LLMs using DYVAL across mathematics, logical reasoning, and algorithmic problems, highlighting the importance of dynamic evaluation. They also demonstrate the effectiveness of DYVAL-generated data in fine-tuning LLMs on existing benchmarks. Key findings include inconsistencies between DYVAL and existing benchmarks, LLMs' performance decline with increasing complexity, and insights into failure patterns and prompt engineering methods.

**Strengths:**

1. DYVAL presents an innovative approach to evaluating LLMs by dynamically generating evaluation samples, mitigating concerns about data contamination and providing a more realistic assessment of LLMs' capabilities.

2. The paper conducts extensive experiments across various reasoning tasks and LLMs, offering valuable insights into LLM performance, failure patterns, and the impact of different prompt engineering methods.

3. DYVAL's ability to improve LLMs' performance on existing benchmarks through fine-tuning with DYVAL-generated data demonstrates its practical utility in enhancing LLM capabilities beyond evaluation

**Weaknesses:**

1. The claim on "co-evolution" is not clear. I do not quite understand what co-evolution means. It seems that the evaluation process is not dependent on the LLM, then how they are correlated from each other.

2. The data contamination problem is not clear. Notably, the data generated by the proposed method is rather limited type as it can not generate narrative generation tasks and others related to common sense.  I am wondering how the existing datasets have the contamination problem. I think such a problem may not happen frequently in the logical reasoning and algorithm domains (especially, these abilities may be majorly from finetune from code and scientific papers). However, they are much easier to happen on those storytelling data.

3. The potential bias may exist in the graph generation. The paper focuses on how to conduct constraints for the graph to avoid illegal ones. Nonetheless, there may be lacked of details on how the graph is generated to meet those constraints. I am concerned that the graph generation algorithms remain biased. Therefore, there will be bias in the generated text, leading to the potential issue.

**Questions:**

1. Can you clarify the concept of "co-evolution" in the context of DYVAL's evaluation process?

2. Could you show the data contamination problem in existing datasets on the proposed problems?

3. Is there a risk of potential bias in the graph generation process that could lead to biased text generation?

---

> ### Author Response · Authors · 2023-11-15
> **Response to Review ydTt (Part 1)**
>
> ### Explain "co-evolution"
>
> We apologize for any misunderstanding of the term "co-evolution". By this term, we mean that evaluation protocols should also evolve to be more complex to test the rapid development of LLMs. We noticed that current benchmarks remain **static**, i.e., most of them will not be updated once they are released to public; while it is the fact the LLMs do keep evolving (the network becomes larger, with more training data). So, the dilemma happens: LLMs are becoming stronger while evaluation benchmarks stays static, leading to an unfair evaluation.
>
> Therefore, we design DyVal to be not only a dynamic benchmark, but also a **flexible** one that can evolve with the development of LLMs. For example, as shown in the following table, both ChatGPT and GPT-4 achieved 100% accuracy in math and logic reasoning tasks in BigBench. It can also be observed that in our experiments: in the simplest arithmetic and boolean logic tasks, the performance gap between GPT-4 and ChatGPT is marginal. Yet, as the complexity of tasks increases, this gap becomes increasingly pronounced. In our framework, DyVal can evolve the evaluation benchmarks by increasing the complexity of tasks, preventing them from plateauing in performance. As shown in Sec. 4.2, the performance of ChatGPT significantly drops as tasks becoming more challenging. This observation validates the effectiveness of DyVal in providing a evolving test bed for LLMs.
>
> ### Present data contamination in existing benchmarks
> >The data contamination problem is not clear. Notably, the data generated by the proposed method is rather limited type as it can not generate narrative generation tasks and others related to common sense. I am wondering how the existing datasets have the contamination problem. I think such a problem may not happen frequently in the logical reasoning and algorithm domains (especially, these abilities may be majorly from finetune from code and scientific papers). However, they are much easier to happen on those storytelling data.
>
> We answered this question in the general response about data contamination. Now we give more references to show how serious this problem is.
>
> There are already several studies [1,2,3,4,5,6,7,8] (before and after our work) demonstrated the data contamination issue. Further, our observations reveal a notable performance gap between LLMs like phi-1.5 [9], Xwin-13B [10], wizard-math [11] on standard benchmarks (e.g., GSM8K [12]) and the arithmetic tasks generated by our DyVal framework. Despite their reported high performance on  GSM8K, these models exhibit significantly lower efficiency on DyVal tasks. This discrepancy points towards potential data contamination in existing benchmarks, where LLMs might have been inadvertently trained on test set data, leading to artificially inflated performance. As outlined in Section 3.2.1, DyVal's dynamic generation process is designed to prevent any overlap with training datasets, thereby providing a more accurate assessment of LLM capabilities.
>
> Finally, existing evidence has shown that the logical and math problems are also having the contamination problem (e.g., the training data of phi-1.5 contains a lot of such reasoning tasks). This shows the necessity of our work.
> Moreover, we do not limit ourselves in reasoning tasks only. As shown in the general response, our framework is flexible to other NLP tasks, which may undergoes more serious data contamination.
>
> [1] GPT-4 performs significantly worse on coding problems not in its training data. https://brianlovin.com/hn/35297067
>
> [2] No, GPT4 can’t ace MIT. https://flower-nutria-41d.notion.site/No-GPT4-can-t-ace-MIT-b27e6796ab5a48368127a98216c76864
>
> [3] Kocoń, Jan, et al. "ChatGPT: Jack of all trades, master of none." Information Fusion (2023): 101861.
>
> [4] Zečević, Matej, et al. "Causal parrots: Large language models may talk causality but are not causal." Transactions on Machine Learning Research 2023.
>
> [5] Wei, Tianwen, et al. "Skywork: A More Open Bilingual Foundation Model." arXiv preprint arXiv:2310.19341 (2023).
>
> [6] Zhou, Kun, et al. "Don't Make Your LLM an Evaluation Benchmark Cheater." arXiv preprint arXiv:2311.01964 (2023).
>
> [7] Li, Yucheng. "An Open Source Data Contamination Report for Llama Series Models." arXiv preprint arXiv:2310.17589 (2023).
>
> [8] Golchin, Shahriar, et al. "Data Contamination Quiz: A Tool to Detect and Estimate Contamination in Large Language Models." arXiv preprint arxiv:2311.06233 (2023).
>
> [9] Li, Yuanzhi, et al. "Textbooks are all you need ii: phi-1.5 technical report." arXiv preprint arXiv:2309.05463 (2023).
>
> [10] Luo, Haipeng, et al. "Wizardmath: Empowering mathematical reasoning for large language models via reinforced evol-instruct." arXiv preprint arXiv:2308.09583 (2023).
>
> [11] Xwin-LM. https://github.com/Xwin-LM/Xwin-LM
>
> [12] Cobbe, Karl, et al. "Training verifiers to solve math word problems." arXiv preprint arXiv:2110.14168 (2021).

---

> ### Author Response · Authors · 2023-11-15
> **Response to Review ydTt (Part 2)**
>
> ### Lacked of details for constraint construction and potential bias in evaluation
> > The potential bias may exist in the graph generation. The paper focuses on how to conduct constraints for the graph to avoid illegal ones. Nonetheless, there may be lacked of details on how the graph is generated to meet those constraints. I am concerned that the graph generation algorithms remain biased. Therefore, there will be bias in the generated text, leading to the potential issue.
>
>
> Thanks for your valuable concerns. We have submitted the code in the revision. As an example, to prevent illegal operations like division by zero in arithmetic tasks, we implement a detection and regeneration mechanism. If a node representing divident, e.g., 'C' in a division operation 'A=B/C', is initially zero, it is automatically regenerated until a non-zero value is obtained.
>
> ```
>
>    A(\)                          A(\)
>    / \      -> Regenerate C      / \
> B(3) C(0)                     B(3) C(1)
> ```
>
>
> We would greatly appreciate further details about the "potential bias". If the potential biases here refers to issues of dataset imbalance, as pointed by Reviewer iwJv, our algorithm can easily satisfy the balance requirement by meticulously controlling the flexible dynamic generation process. For example, in reachability task, we can drop the generated evaluation samples with 'False' labels until we generate a sample with 'True' label. In fact, we presented the **results of ChatGPT and GPT4 in balanced datasets** in Table 1. The results in balanced datasets are similar to our initial findings: (1) ChatGPT consistently predicted all questions as "True.", resulted in a uniform accuracy rate of 50%. (2) GPT-4 demonstrated excellent performance. It maintained significantly higher accuracy rates across all complexity levels.
>
> | **Model**      | **ChatGPT** |   |   |   | **GPT4** |   |   |   |
> |----------------|-------------|-------|-------|-------|----------|-------|-------|-------|
> | **Complexity** | D1          | D2    | D3    | D4    | D1       | D2    | D3    | D4    |
> | **Balanced**   | 50          | 50.00 | 50.00 | 50.00 | 84.54    | 79.03 | 73.50 | 72.41 |
> | **Imbalanced** | 63.87       | 54.27 | 53.73 | 52.33 | 85.33    | 83.00 | 67.67 | 74.33 |

---

> > ### Comment · Reviewer_ydTt · 2023-11-18
> > **Thanks for response**
> >
> > Thanks for your response. I raise my score correspondingly. Nonetheless, I hope the author will further improve the paper's quality with the following instructions: (1) Have a more comprehensive comparison with other synthetic datasets following the same logic as this method. (2) provide a comprehensive code framework (3) add your explanation for me in the revision since I believe those words are inexact expressions leading to confusion.

---

> > > ### Author Response · Authors · 2023-11-22
> > >
> > > Thank you for the recognition of our work. Indeed, we will keep polishing the paper following your constructive instructions! The discussions in the response are included in the revision. And the initial code is provided in the supplementary zone while we are working on making a more comprehensive codebase to compare with other synthetic datasets. In the future, we will add those comparisons in the new version of the paper once these experiments are done. Thank you for your comments in making this paper better:)

---

### Official Review · Reviewer_MYvY · 2023-10-30

**Soundness:** 3 good
**Presentation:** 3 good
**Contribution:** 2 fair
**Rating:** 6
**Confidence:** 5

**Summary:**

The paper proposes a new dynamic generation of samples that can be used to evaluate or fine-tune LLMs. Roughly speaking, a sample corresponds to a DAG with controllable complexity that can be translated into a comprehensible natural language description. This translated sample and a task description can then form an evaluation task for the LLM. The proposal is to use dynamic draws of graph-informed samples to evaluate LLMs and potentially train and fine-tune them on specific tasks. Since the space of large DAGs is exponentially large, it's very unlikely to observe repetitive samples, and hence the algorithm addresses two potential flaws of static benchmarks: data contamination and saturation due to static complexity.

**Strengths:**

- Extensive experiments are conducted.
- Graph-based notions of complexities can be used as a means to control the compositional complexity of the examples.
- Address data contamination and static complexity of the benchmarks.

**Weaknesses:**

- A common challenge associated with this framework is the need to manually specify a problem as a computation graph with valid constraints. This requirement is only understandable if LLM is intended to acquire specific skills written in these formats.
- Before reading this paper, I believed that generating a large number of mathematical problems of specific types and evaluating LLMs on them was primarily for debugging specific LLM capabilities, such as compositionality, rather than as an evaluation framework. I'm not sure if these types of problems are fundamental questions about LLMs. In fact, prior studies, such as those by Dziri et al., have already highlighted the limitations of transformers in these settings, using a very similar setup for demonstration.
- It's not clear if LLMs are losing some skills when fine-tuned on DyVal as DyVal examples and the chosen existing benchmarks are from very similar domains. The generalization of the fine-tuned model on DP is interesting though.

Recommendation:
As a person who has worked on dynamic adversarial data collection, or more broadly dynamic benchmarks, I think your review of this literature is underestimating their importance. In fact, in dynamic adversarial data collection annotators can be provide interesting problem instances hard to find in static benchmarks and even hard to manually specify as a DyVal task. So, I encourage you to include a better review of these works. If you are concerned with the human-in-the-loop, I believe the recent theoretical frameworks of dynamic benchmarking are still valid if humans are replaced by generative tools which you may consider mentioning. So, I encourage you to revisit page 2 paragraph 1 at your discretion.

Overall, I believe that in the era of LLMs, we should explore new methods of evaluation, and this paper's framework might be one of them. The ICLR audience may find this work interesting, so I will maintain a positive rating despite the concerns I have.

**Questions:**

Feel free to respond to the weaknesses.

---

> ### Author Response · Authors · 2023-11-15
> **Response to Review MYvY (Part 1)**
>
> ### Limited application scenary
> > A common challenge associated with this framework is the need to manually specify a problem as a computation graph with valid constraints. This requirement is only understandable if LLM is intended to acquire specific skills written in these formats.
>
> This relates to the importance of reasoning tasks in LLM and the flexibility of our approach. We have answered this question in the general response. Now we briefly answer it.
> Reasoning ability is widely recognized as the core of both human and artificial intelligence, our focus on constructing reasoning tasks mirrors the intricate and multi-step nature of human reasoning [1, 2, 3], building reasoning benchmarks is a crutial step to help LLMs towards intelligence.
> Furthermore, our proposed dynamic evaluation protocol, including graphs and trees, can also be applied to other tasks such as natural language understanding (NLU) tasks. Take sentiment analysis as an example, We can abstract a sentence into a syntax tree and form tree templates which can be used to generate diverse sentences. Take 'I am happy today' as an instance, it can be formalized as:
>
> ```
>      Subject (I)
>          |
>      Predicate
>     /        \
> Verb (am)   Complement
>             /        \
>   Adjective (happy)  Temporal Phrase (today)
> ```
>
>
> Following this template, we can create variations by replacing the corresponding words, like 'He was sad yesterday', 'She is excited now'.
>
> Backing to the reasoning tasks, it is reasonable to solve a arithmetic, or logical reasoning problem in the daily interactions with LLMs. And our generated samples, although a little bit straightforward, are similar to those real-world benchmarks. Here is an example of arithmetic problem:
>
> ```
> Here is a description of an arithmetic problem:
>
> The value of aaa is 9.
> The value of aad is 4.
> aae gets its value by taking the square root of the value that aad has.
> The value of aab is 3.
> aac gets its value by adding together the value of aaa and aab.
> aaf gets its value by subtracting the value of aae from the value of aac.
>
> Compute the result of aaf.
> ```
>
> [1] Brody, Nathan. "What is intelligence?." International Review of Psychiatry 11.1 (1999): 19-25.
>
> [2] Lohman, D., & Lakin, J. (2011). Intelligence and Reasoning. In R. Sternberg & S. Kaufman (Eds.), The Cambridge Handbook of Intelligence (Cambridge Handbooks in Psychology, pp. 419-441).
>
> [2] Sawada, Tomohiro, et al. "Arb: Advanced reasoning benchmark for large language models." arXiv preprint arXiv:2307.13692 (2023).
>
>
> ### Difference between 'faith and fate' and 'DyVal'
> > Before reading this paper, I believed that generating a large number of mathematical problems of specific types and evaluating LLMs on them was primarily for debugging specific LLM capabilities, such as compositionality, rather than as an evaluation framework. I'm not sure if these types of problems are fundamental questions about LLMs. In fact, prior studies, such as those by Dziri et al. (Faith and Fate: Limits of Transformers on Compositionality), have already highlighted the limitations of transformers in these settings, using a very similar setup for demonstration.
>
> - First of all, mathematical problems are indeed a fundamental evaluation problem in LLMs, as shown in GPT-4 report and many other benchmarks. We refer the reviewer to general response for more explanation of reasoning tasks.
> - Second, while the "faith and fate" paper claimed Transformer architecture faces limits in compositionality, current LLM benchmarks are still relying on the evaluation of math problems to show their strength. So, such evaluation is still valid.
> - Finally and most importantly, the goal of DyVal is totally different with the computational graphs analyzed in Dziri et al.' work. DyVal aims to mitigate the data contamination and static nature of current benchmarks. While Dziri et al.' work indicated that LLMs learned spurious correlation and tried to match subgraphs during reasoning process.

---

> ### Author Response · Authors · 2023-11-15
> **Response to Review MYvY (Part 2)**
>
> ### General ability after fine-tuning
> > It's not clear if LLMs are losing some skills when fine-tuned on DyVal as DyVal examples and the chosen existing benchmarks are from very similar domains. The generalization of the fine-tuned model on DP is interesting though.
>
> Thanks for your valuable suggestions. First of all, there is no unanimous option in the community about the ability before and after fine-tuning: they could be stronger on some tasks and weaker on other tasks. This is still an open question. Therefore, we do not have a concrete answer in all models and tasks to this question.
>
> To resolve your concerns, we fine-tuned GPT-3.5-turbo-0613 using our generated data on abductive logic and reachability datasets, as GPT-3.5 performs worst on these two datasets. Specifically, we generated 100 samples across complexity levels D1, D2, and D3 for each task. We compared the performance of original and fine-tuned models on several benchmark tasks in GLUE dataset. The performance of abductive logic and reachability task are tested on D4 task (different from fine-tuning dataset). As shown below, performance on WNLI and QNLI datasets dropped for the fine-tuned model. However, fine-tuned model achieves better results on CoLA, QQP, and MRPC datasets. Despite the mixed results, the overall improvement in several datasets suggests that fine-tuning on our generated datasets *does not* necessarily hurt the general language understanding ability.
>
> |  | Abductive Logic | Reachability | SST-2 | CoLA | WNLI | QNLI | QQP | MRPC |
> |------------------------------|----------------------------------------------|-------------------------------------------|------------------------------------|-----------------------------------|-----------------------------------|-----------------------------------|----------------------------------|-----------------------------------|
> | GPT-3.5                       | 55.27                                        | 50.00                                     | **93.29**                              | 77.00                             | **59.15**                             | **80.00**                             | 76.50                            | 73.0                              |
> | GPT-3.5-FT                    | **85.10**                                        | **96.53**                                     | 93.23                              | **78.00**                             | 45.07                             | 72.50                             | **78.00**                            | **77.5**                              |

---

> > ### Comment · Reviewer_MYvY · 2023-11-15
> >
> > Thank you for providing further clarification.
> >
> > The mixed results after fine-tuning are interesting, and I suggest including them in the revised paper.
> >
> > I understand the difference between faith and fate and DyVal, but thank you for the clarification.
> >
> > I had no doubt about the importance of the reasoning task, but I have concerns about the flexibility of the method to generate natural examples. Sentiment analysis is a good example. I can see that many natural language examples follow a hierarchical structure, a syntax tree. This means natural language examples typically can be specified by a DAG. However, when it comes to generating new graphs, we need to specify constraints, which might encompass the entire English grammar. It's also not clear whether the generated examples will be truly natural. Another factor that can limit the complexity of generated graphs is that in sentiment analysis, if the sentence is long or the DAG is large, the true sentiment might not be clear. Therefore, we may need to adhere to a few templates for which the generated sentences will have a clear sentiment.
> >
> > Overall, thank you for your response. I would like to maintain my current rating. This is an area of great interest, and this work can be of interest to ICLR's audience, although I have some doubts about the flexibility of the framework to generate natural examples.

---

> > > ### Author Response · Authors · 2023-11-16
> > > **Further response to Reviewer MYvY**
> > >
> > > Thank you for your insightful feedback. Our current DAG-based implementation of dynamic evaluation is just the beginning to introduce the graph-based generation to evaluation of LLMs. We focus on reasoning tasks in this paper. In the following, we show that DyVal is flexible to support general NLP tasks. But as we stated in the conclusion of the paper, this should be left for future work since it is nontrivial.
> > >
> > > > if the sentence is long or the DAG is large, the true sentiment might not be clear.
> > >
> > > Yes, this is a challenging point when using syntax tree templates to generate sentences. However, we believe it can be mitigated by leveraging LLMs, as most textual adversarial attacks employed. For example, BertAttack utilizes BERT model to replace words while maintaining semantic integrity.
> > >
> > > Here, we implement the semantic syntax tree templates for SST-2 tasl. Specifically, for each sentence in the SST-2 dataset, we initially employ ChatGPT to extract its syntactic structure. Within each syntax tree, we identify the elements that can be modified: **nouns (such as names and places) and adjectives**. ChatGPT is then used to create five alternative **candidates** for each of these modifiable components, which are subsequently replaced in an iterative fashion. Throughout this process, we continuously assess whether these replacements alter the original **semantic meaning** of the sentence. Any changes that result in a semantic shift are discarded.
> > >
> > > We generate three alternative versions for each sentence in the above process, forming our newly generated dataset. We then evaluate the performance of both Flan-T5-large and Llama2-7b models, using the original SST-2 dataset as well as our generated dataset for comparison. The results of these evaluations are detailed in the following table.
> > >
> > > |        | Flan-T5-large | Llama2-7b |
> > > |--------|---------------|-----------|
> > > | Origin | 93.12         | 90.37     |
> > > | DyVal  | 87.46         | 72.03     |

---

> > > > ### Comment · Reviewer_MYvY · 2023-11-16
> > > >
> > > > I appreciate your effort and the new experiment. I have already suggested acceptance and this made me more confident (I updated my confidence score).

---

### Official Review · Reviewer_sT7J · 2023-10-31

**Soundness:** 3 good
**Presentation:** 3 good
**Contribution:** 3 good
**Rating:** 6
**Confidence:** 4

**Summary:**

Evaluating LLMs is important in current literature as LLM has boosted significant performance in various tasks. This paper proposes a new evaluation method that evaluates the performance of various LLMs in their reasoning abilities by generating dynamic evaluation samples. Results show that several tasks are still hard for current LLMs.

**Strengths:**

1. The motivation of this paper is clear. As many LLMs tend to memorize static data for evaluation, this paper proposes a dynamic approach to avoid this kind of problem.

2. The idea of generating tasks with different difficulties in a DAG style sounds interesting.

3. The problem is clearly described with sufficient notations and examples.

4. Experiments are conducted in various aspects, including 7 reasoning tasks, 1 human evaluation, on about 8 well-known LLMs. Fine-tuning experiments are also conducted to demonstrate that the LLMs' ability in learning to reason.

**Weaknesses:**

1. The title is somewhat misleading. The evaluation tasks in this paper are mostly about reasoning on maths, logic, algorithms, etc. However, the title reflects no information about this point. The abstract could be also clearer if this point can be mentioned earlier.

2. For the fine-tuning results in Section 5, I wonder when these LLMs are fine-tuned for the reasoning tasks proposed in this method, will the general abilities be influenced? Or to what extent will they be influenced?

3. As the samples for evaluation are dynamic, the comparison may be unfair when the generated data are different in different evaluation stages.

**Questions:**

1. Can you discuss the influence of fine-tuning on reasoning tasks on the general language understanding ability?

2. Can you provide how to fairly evaluate the different models, especially if this evaluation method is released as a public leaderboard?

---

> ### Author Response · Authors · 2023-11-15
> **Response to Review sT7J**
>
> ### Misleading title
> > The evaluation tasks in this paper are mostly about reasoning on maths, logic, algorithms, etc. However, the title reflects no information about this point. The abstract could be also clearer if this point can be mentioned earlier.
>
> Thank you for your suggestions. We have revised the title to better reflect the focus on reasoning tasks. However, we would like to highlight that our proposed dynamic evaluation protocol can go beyond reasoning tasks in this paper to other tasks such as natural language understanding (NLU) tasks. Take sentiment analysis as an example, We can abstract a sentence into a syntax tree and form tree templates which can be used to generate diverse sentences. Take 'I am happy today' as an instance, it can be formalized as:
>
> ```
>      Subject (I)
>          |
>      Predicate
>     /        \
> Verb (am)   Complement
>             /        \
>   Adjective (happy)  Temporal Phrase (today)
> ```
>
>
> Following this template, we can create variations by replacing the corresponding words, like 'He was sad yesterday', 'She is excited now'.
>
> For more information, please refer to the general response.
>
> ### General ability after fine-tuning
> > I wonder when these LLMs are fine-tuned for the reasoning tasks proposed in this method, will the general abilities be influenced? Or to what extent will they be influenced? Can you discuss the influence of fine-tuning on reasoning tasks on the general language understanding ability?
>
> Thanks for your valuable suggestions. First of all, there is no unanimous option in the community about the ability before and after fine-tuning: they could be stronger on some tasks and weaker on other tasks. This is still an open question. Therefore, we do not have a concrete answer in all models and tasks to this question.
>
> To resolve your concerns, we fine-tuned GPT-3.5-turbo-0613 using our generated data on abductive logic and reachability datasets, as GPT-3.5 performs worst on these two datasets. Specifically, we generated 100 samples across complexity levels D1, D2, and D3 for each task. We compared the performance of original and fine-tuned models on several benchmark tasks in GLUE dataset. The performance of abductive logic and reachability task are tested on D4 task (different from fine-tuning dataset). As shown below, performance on WNLI and QNLI datasets dropped for the fine-tuned model. However, fine-tuned model achieves better results on CoLA, QQP, and MRPC datasets. Despite the mixed results, the overall improvement in several datasets suggests that fine-tuning on our generated datasets *does not* necessarily hurt the general language understanding ability.
>
> |  | Abductive Logic | Reachability | SST-2 | CoLA | WNLI | QNLI | QQP | MRPC |
> |------------------------------|----------------------------------------------|-------------------------------------------|------------------------------------|-----------------------------------|-----------------------------------|-----------------------------------|----------------------------------|-----------------------------------|
> | GPT-3.5 | 55.27  | 50.00 | **93.29**  | 77.00 | **59.15**   | **80.00**   | 76.50 | 73.0  |
> | GPT-3.5-FT                    | **85.10**                                        | **96.53**                                     | 93.23                              | **78.00**                             | 45.07 | 72.50 | **78.00**                            | **77.5**                              |
>
>
> ### Unfair evaluation among LLMs
> > As the samples for evaluation are dynamic, the comparison may be unfair when the generated data are different in different evaluation stages. Can you provide how to fairly evaluate the different models, especially if this evaluation method is released as a public leaderboard?
>
> Thank you for this valuable insight. Indeed, unfair comparison could happen without care. In our work, we have taken careful measures to ensure that our evaluation is fair and consistent across different LLMs.
> - First, the generation mechanism indicates that there is no same evaluation sets for each algorithm. That's why we need to introduce randomness and multiple runs in the experiments. We never said that the results in a single run can be used to reflect the performance, but on multiple runs.
> - Specifically, our experiments are done in multiple runs to eliminate such bias and unfairness. For each level of complexity (denoted as D1 to D4 in our paper, with D1 being the simplest), we generate a set of 500 samples. We run each evaluation *three* times. We present the mean and standard deviation of the performance metrics in Appendix Tables 4, 5, and 6. The standard deviations are *minor*, typically ranging between 0.5 to 1.5.
> - Finally, we acknowledge there is always room for fairness enhancement such as increasing the number of evaluation samples and the frequency of repetitions can further refine the accuracy and fairness of our assessments.

---

> > ### Comment · Reviewer_sT7J · 2023-11-17
> >
> > Thank you for the detailed feedback. Some of my concerns are addressed. However, I find that almost no revision is done in the new version, except for the title and an extra section in the appendix. For example, several reviewers have concerns regarding the importance of reasoning task and discussion on related works. I suggest the authors further adjust related contents in the paper to ease these weak points and make the contents clearer for a wider range of readers.

---

> > > ### Author Response · Authors · 2023-11-22
> > >
> > > Thanks for the further comments! We have uploaded a new revision to include all the discussions (marked blue) as suggested. Please do not hesitate to let us know if there's any new question:)

---

### Official Review · Reviewer_iwJv · 2023-11-03

**Soundness:** 3 good
**Presentation:** 3 good
**Contribution:** 3 good
**Rating:** 6
**Confidence:** 3

**Summary:**

Presents a general framework to generate certain "graph-based" evaluation tasks for LLMs randomly, implements 7 example tasks, and presents and analyzes empirical results.

**Strengths:**

S1. Simple, yet flexible framework.
S2. Dynamic task generation with controllable complexity
S3. Extensive evaluation of selected LLMs / prompting strategies for seven simple reasoning tasks.

On S1. The general idea of the proposed benchmarking framework is to generate tasks that can be described by a directed acyclic graph. This includes "compute graphs" (e.g., evaluate a numerical expression or perform logical reasoning) or "data graphs" (e.g., determine connectivity between vertices). The framework takes care of graph generation, task implementations add contraints, labels, solutions, and verbalization. This is a very natural approach and (most probably) how many of the existing benchmarks of this form are generated in the first place. Such a framework may increase usability, especially when many tasks were implemented in it.

On S2. Tasks are generated automatically and with varying complexity (mainly graph size). Again, this is a simple, very natural approach. Here the framework proposed by the paper may make comparative evaluation across a range of tasks more feasible, as all share the same notion of "complexity".

On S3. The paper reports performance results on simple computational tasks (such as evaluating simple equations). Generally, all models break down when complexity goes up so that the benchmark may be used as a way to evaluate progress. Also, the performance reported on these simple tasks sometimes contradict performance results published on related, static benchmarks.

**Weaknesses:**

W1. Certain computational tasks only
W2. Discussion of related work / results lacking
W3. Limitations in generated graphs
W4. Code/data availability unclear
W5. Limited insight of experimental study

On W1. By the nature of the benchmark, it focuses on problems that can be expressed as (currently small) compute graphs or data graphs and are somewhat artificial. It only tests a very limited field of LLM functionality.

On W2. There are benchmarks for all of the tasks that are implemented in this framework already. The paper states that its performance results contradict the ones on some of these benchmarks, but does not say which ones and, perhaps more importantly, does not provide any insight into why this is the case. Also, the data generation strategies used by existing benchmarks are not discussed. Finally, to what extent the benchmark can be used to really do new things (beyond existing benchmarks) is not discussed.

On W3. First, the paper focuses solely on DAGs, but it's unclear why this is done for data graphs (e.g., reachability, max-sum). Second, it's unclear whether graph size is the right complexity measure. E.g., for reachability appears easier is source and target are neighbors, no matter how large the graph. Finally, the system does not seem to generate balanced datasets. For example, the paper reports in the appendix that the proportion of true answers for reachability is not controlled, leading to "paradoxical" results.

On W4. It's important for benchmarking papers such as this one to make all code, datasets, prompts, results, etc. public. The paper currently does not provide any ressources (or, at least, I did not see them).

On W5. The insight that can be drawn from the experiments is somewhat limited. I do not count this against the paper, however. It does show exposed limitations of LLMs and prompting strategies, and it does show that the generated tasks are useful for fine-tuning.

Minor points:

I am not sure how useful the comparison to human performance is. Clearly, all of the tasks can be solved "easily" by humans, it's just a pain to do so.

**Questions:**

None

---

> ### Author Response · Authors · 2023-11-15
> **Response to Review iwJv (Part 1)**
>
> ### Why reasoning tasks?
>
> > By the nature of the benchmark, it focuses on problems that can be expressed as (currently small) compute graphs or data graphs and are somewhat artificial. It only tests a very limited field of LLM functionality.
>
> Thank you for your valuable review. The importance of reasoning tasks in LLM evaluation can be found in "general response" section. Here we emphasize the flexibility of our DyVal framework by constructing a natural language understanding task: Take sentiment analysis as an example, We can abstract a sentence into a syntax tree and form tree templates which can be used to generate diverse sentences. Take 'I am happy today' as an instance, it can be formalized as:
>
> ```
>      Subject (I)
>          |
>      Predicate
>     /        \
> Verb (am)   Complement
>             /        \
>   Adjective (happy)  Temporal Phrase (today)
> ```
>
> Following this template, we can create variations by replacing the corresponding words, like 'He was sad yesterday', 'She is excited now'.
>
>
>
> ### Code/data availability
> We have submitted our code in the revision.
>
>
> ### Limited insight of experimental study
> > The insight that can be drawn from the experiments is somewhat limited. I do not count this against the paper, however. It does show exposed limitations of LLMs and prompting strategies, and it does show that the generated tasks are useful for fine-tuning.
>
> With all due respect, we *cannot* agree with this comment saying that we provide "limited insight" in this study. We would like the reviewer to refer to the general response to understand the importance of this work in fighting against the data contamination issue: we are the first dynamic algorithm for LLM evaluation with some interesting findings that were *never* demonstrated before.
> - First of all, our primary goal is not merely to introduce another challenging benchmark. Instead, we aim to establish a dynamic evaluation protocol trying to address two critical issues in evaluating LLMs: data contamination and the static nature of current benchmarks. By dynamically generating and evolving our evaluation tasks, we intend to present a more accurate and robust method for assessing LLMs' true capabilities.
> - Second, our experimental results demonstrate some **new** findings that were never revealed before:
>     - Data contamination issue: There exists a significant discrepancy in the performance of some LLMs on static benchmarks versus our dynamic DyVal benchmarks. This contrast provides strong evidence of the data contamination issue in existing benchmarks, where models may inadvertently be trained on test data, thus skewing their performance.
>     - Efficacy of DyVal as an evolving benchmark: Our results also demonstrate the potential of DyVal as an evolving benchmark. It successfully challenges LLMs with tasks that increase in complexity, ensuring that the models can be continuously tested against more advanced and varied problems.
>     - Prompt engineering can help in certain tasks, but still fails in our benchmarks.
>
> We do hope the reviewer can have a better understanding of our contributions and findings.

---

> ### Author Response · Authors · 2023-11-15
> **Response to Review iwJv (Part 2)**
>
> ### Discussion of related work / results lacking
> > There are benchmarks for all of the tasks that are implemented in this framework already. The paper states that its performance results contradict the ones on some of these benchmarks, but does not say which ones and, perhaps more importantly, does not provide any insight into why this is the case. Also, the data generation strategies used by existing benchmarks are not discussed. Finally, to what extent the benchmark can be used to really do new things (beyond existing benchmarks) is not discussed.
>
> Thank you for your valuable feedback. We acknowledge the presence of existing benchmarks and the need to clarify how our work differs and contributes beyond them. In response to your concerns, we offer the following insights:
>
> 1. **Data contamination issue:** As we discussed in related work, current benchmarks are mostly public collected datasets from Internet, DynaX series using crowd-source to generate dynamic dataset. We observed a remarkable discrepancy between the performance of certain Large Language Models (LLMs), such as phi-1.5 [1], wizard-math [2] and Xwin-13B [3] on widely-used benchmarks such as GSM8K [4], and their performance on the math tasks generated by our framework. These models show nearly zero efficiency on our tasks, despite claiming comparable (even state-of-the-art) performance on GSM8K. This discrepancy suggests a severe data contamination issue [5,6,7,8,9,10,11,12]. LLMs might be indirectly trained on parts of these test sets, inflating their genuine performance. As mentioned in Sec. 3.2.1, the dynamic generation process of DyVal ensures no overlap with training datasets, guarantee robust and accurate evaluation.
>
>     [1] Li, Yuanzhi, et al. "Textbooks are all you need ii: phi-1.5 technical report." arXiv preprint arXiv:2309.05463 (2023).
>
>     [2] Luo, Haipeng, et al. "Wizardmath: Empowering mathematical reasoning for large language models via reinforced evol-instruct." arXiv preprint arXiv:2308.09583 (2023).
>
>     [3] Xwin-LM. https://github.com/Xwin-LM/Xwin-LM
>
>     [4] Cobbe, Karl, et al. "Training verifiers to solve math word problems." arXiv preprint arXiv:2110.14168 (2021).
>
>     [5] GPT-4 performs significantly worse on coding problems not in its training data. https://brianlovin.com/hn/35297067
>
>     [6] No, GPT4 can’t ace MIT. https://flower-nutria-41d.notion.site/No-GPT4-can-t-ace-MIT-b27e6796ab5a48368127a98216c76864
>
>     [7] Kocoń, Jan, et al. "ChatGPT: Jack of all trades, master of none." Information Fusion (2023): 101861.
>
>     [8] Zečević, Matej, et al. "Causal parrots: Large language models may talk causality but are not causal." arXiv preprint arXiv:2308.13067 (2023).
>
>     [9] Wei, Tianwen, et al. "Skywork: A More Open Bilingual Foundation Model." arXiv preprint arXiv:2310.19341 (2023).
>
>     [10] Zhou, Kun, et al. "Don't Make Your LLM an Evaluation Benchmark Cheater." arXiv preprint arXiv:2311.01964 (2023).
>
>     [11] Golchin, Shahriar, and Mihai Surdeanu. "Time travel in llms: Tracing data contamination in large language models." arXiv preprint arXiv:2308.08493 (2023).
>
>     [12] Yang, Shuo, et al. "Rethinking Benchmark and Contamination for Language Models with Rephrased Samples." arXiv preprint arXiv:2311.04850 (2023).
>
> 2. **Static nature of current benchmarks:** For advanced LLMs such as ChatGPT and GPT-4 , they tend to achieve near-perfect scores on static benchmarks, rendering these evaluations less effective for measuring progress. For example, as shown in the following table, both ChatGPT and GPT-4 achieved 100% accuracy in math and logic reasoning tasks in BigBench. In our framework, DyVal can evolve the evaluation benchmarks by increasing the complexity of tasks, preventing them from plateauing in performance. As shown in Sec. 4.2, the performance of ChatGPT significantly drops as tasks becoming more challenging. This observation validates the effectiveness of DyVal in providing a evolving testbed for LLMs.
>
>
>
> > To what extent the propose benchmark can do new things?
>
> This is the main contribution of our benchmark: dynamic generation of testing samples to overcome the data contamination issue, which is completely new. This also relates to the importance of this research, and the reviewer is suggested to refer to the general response section.

---

> ### Author Response · Authors · 2023-11-15
> **Response to Review iwJv (Part 3)**
>
> ### Limitations in generated graphs for algorithm tasks
> > First, the paper focuses solely on DAGs, but it's unclear why this is done for data graphs (e.g., reachability, max-sum). Second, it's unclear whether graph size is the right complexity measure. E.g., for reachability appears easier is source and target are neighbors, no matter how large the graph. Finally, the system does not seem to generate balanced datasets. For example, the paper reports in the appendix that the proportion of true answers for reachability is not controlled, leading to "paradoxical" results.
>
> - > Why focusing on DAGs?
> 	- DAG is a fundamental structure in computer science used to model a wide range of problems where entities (like cities) are represented as nodes and connections as links. The decision to focus on DAGs was based on their prevalence and applicability in real-world scenarios. For instance, they are crucial in scenarios like project scheduling, dependency resolution, and more, thus providing a practical context for evaluating LLMs.
> - > Unclear why this is done for data graphs
> 	- Reachability and Max-sum are fundamental tasks in graph algorithms, each with significant applicability in various real-world contexts, such as ransportation and network analysis.
>
> - > Why graph size is the right complexity measure?
> 	-  Graph size is generally used in the domain of graph learning to represent certain degree of complexities. Larger graphs require LLMs to efficiently manage and interpret more extensive information, thus challenging their retrieval and search capabilities. The scenario that the source and target are neighbors becomes increasingly less probable as the graph size grows. Therefore, larger graphs still offer a more robust challenge due to the reduced likelihood of neighboring nodes.
>
> - > The system does not seem to generate balanced datasets?
> 	- Our algorithm can easily satisfy the balance requirement. We acknowledge the importance of balanced datasets in avoiding skewed results. This can be easily achieved by meticulously controlling the flexible dynamic generation process. For example, we can drop the generated evaluation samples with 'False' labels until we generate a sample with 'True' label. In fact, we presented the **results of ChatGPT and GPT4 in balanced datasets** in Table 1. The results in balanced datasets are similar to our initial findings: (1) ChatGPT consistently predicted all questions as "True.", resulted in a uniform accuracy rate of 50%. (2) GPT-4 demonstrated excellent performance. It maintained significantly higher accuracy rates across all complexity levels.
>
>         | **Model**      | **ChatGPT** |   |   |   | **GPT4** |   |   |   |
>         |----------------|-------------|-------|-------|-------|----------|-------|-------|-------|
>         | **Complexity** | D1          | D2    | D3    | D4    | D1       | D2    | D3    | D4    |
>         | **Balanced**   | 50          | 50.00 | 50.00 | 50.00 | 84.54    | 79.03 | 73.50 | 72.41 |
>         | **Imbalanced** | 63.87       | 54.27 | 53.73 | 52.33 | 85.33    | 83.00 | 67.67 | 74.33 |

---

> ### Comment · Reviewer_iwJv · 2023-11-15
> **On the response**
>
> Thank you for your response! It's good (and, for this paper, fundamental) that all resources will be made available; I consider W4 addressed. As for my other concerns, most points raised in the response were clear to me already. I'd like to maintain my assessment.
>
> A quick note on data graphs: Here my point was that it is unclear why data graphs are DAGs, instead of general graphs. Also, the complexity of certain graph tasks may not depend on the graph size but on other properties. As a simple example, the shortest path on a "chain graph" takes time linear in the distance of the source and target vertex using classical algorithms, no matter how large the entire graph.
>
> Also note that there exists benchmarks that dynamically generate graphs & tasks, e.g., GraphWorld (https://arxiv.org/abs/2203.00112). This touches on W2 (related work not well covered), a point that has also been raised by other reviewers.

---

> ### Author Response · Authors · 2023-11-15
> **Further response to Reviewer iwJv**
>
> # Further response to Reviewer iwJv
> Thank you for your prompt feedback and for raising important points. We appreciate the opportunity to address these concerns and clarify aspects of our work.
>
> > it is unclear why data graphs are DAGs, instead of general graphs.
>
> We understand the question regarding the choice of DAGs over general graphs. The rationale is rooted in the directional nature of many reasoning tasks. For instance, consider the arithmetic question $(5+3)\times 4$. It naturally formulates a **directed** computation tree, as shown below:
>
> ```
>      E(x)
>     / \
>  D(4)  C(+)
>       / \
>   A(5) B(3)
> ```
> However, we acknowledge that general graphs could also be applicable in certain contexts. Our choice of DAGs is to maintain a consistent framework.
>
> > the complexity of certain graph tasks may not depend on the graph size but on other properties. As a simple example, the shortest path on a "chain graph" takes time linear in the distance of the source and target vertex using classical algorithms, no matter how large the entire graph.
>
> We would like to make the following two clarifications:
> 1. Even for **chain graphs in shortest path task, the complexity still correlates to graph size**. Consider two chain graphs with 5 and 500 nodes. The larger graph inherently requires the processing of a greater number of nodes and an examination of more connections to determine the solution. **This is similar to a longer path requiring more time to traverse, despite its straightforwardness.**
> 2. **Our generation algorithms are designed to ensure graph diversity.** We incorporate randomness in the number of links for each node to create varied graph structures. For instance, in a 20-node graph, each node could have a random number of links ranging from 1 to 7.
>
> > there exists benchmarks that dynamically generate graphs & tasks, e.g., GraphWorld (https://arxiv.org/abs/2203.00112).
>
> Thank you for pointing out the existence of benchmarks like GraphWorld. We will make sure to include a more detailed discussion regarding this in our revised manuscript. However, GraphWorld primarily benchmarks Graph Neural Networks (GNNs), whereas DyVal focuses on benchmarking Large Language Models (LLMs), but just use the graph structure. They are different in nature. DyVal aims to solve two fundamental problems in evaluation of LLMs, as we mentioned above.
>
> - - -
>
> If our response could resolve your concerns, please consider raising the score to support us:) If you have further questions, we are happy to address them.

---

> > ### Comment · Reviewer_iwJv · 2023-11-15
> >
> > As I said, all these points are clear to me. With "data graph", I meant graphs that do not describe structure but data, e.g., the input graphs to the reachability task. The reference to GraphWorld was to emphasize that the discussion of related work is lacking (also, conceptually, the idea of DyVal is to verbalize synthetic graph tasks). That being said, thanks for your inputs, no further clarifications are needed from my side.

---

### Author Response · Authors · 2023-11-15
**General Response**

Thank everyone for pointing out our work is flexible, clear, innovative. We have submitted the code in this revision for reproducibility. As a general response, we would like to highlight several key aspects.

## Importance of this research: data contamination and related work

We would like to highlight the importance of this research: fight against the *data contamination* issue in LLM evaluation, which was actually an active topic these days. Even after the submission of this work (i.e., from October to November), there are several new research efforts [5, 6, 7, 8, 10] deeply highlighting the severity of data contamination:

- Before this work was submitted (before October), we have cited several works in the main paper who expressed concerns of data contamination issue, such as [3, 4, 9]. Apart from these references, there are also some influential blogs expressing the same concern: [1, 2]. In fact, the GPT-4 report and LLama report have clearly stated the phenomenon of data contamination.
- But more support and evidence actually came after this submission (from October to November). Zhou et al. [6] discussed the risks and impacts of data contamination of evaluation benchmarks in assessing LLMs. The Skywork LLM [5] again demonstrated the data contamination issue in several LLMs. The work of [10] designed novel methods to trace the contamination of LLMs. In fact, we feel lucky that we are "somewhat ahead" of them since we are among the first to not only realize, but to propose a new evaluation protocol to fight against this issue. We are also grateful for these works emerging after us for the comprehensive evidence in supporting our motivation. We will cite them in the published version of this paper.

[1] GPT-4 performs significantly worse on coding problems not in its training data. https://brianlovin.com/hn/35297067

[2] No, GPT4 can’t ace MIT. https://flower-nutria-41d.notion.site/No-GPT4-can-t-ace-MIT-b27e6796ab5a48368127a98216c76864

[3] Kocoń, Jan, et al. "ChatGPT: Jack of all trades, master of none." Information Fusion (2023): 101861.

[4] Zečević, Matej, et al. "Causal parrots: Large language models may talk causality but are not causal." TMLR 2023.

[5] Wei, Tianwen, et al. "Skywork: A More Open Bilingual Foundation Model." arXiv preprint arXiv:2310.19341 (2023).

[6] Zhou, Kun, et al. "Don't Make Your LLM an Evaluation Benchmark Cheater." arXiv preprint arXiv:2311.01964 (2023).

[7] Li, Yucheng. "An Open Source Data Contamination Report for Llama Series Models." arXiv preprint arXiv:2310.17589 (2023).

[8] Golchin, Shahriar, and Mihai Surdeanu. "Data Contamination Quiz: A Tool to Detect and Estimate Contamination in Large Language Models." arXiv preprint arxiv:2311.06233 (2023).

[9] Golchin, Shahriar, and Mihai Surdeanu. "Time travel in llms: Tracing data contamination in large language models." arXiv preprint arXiv:2308.08493 (2023).

[10] Yang, Shuo, et al. "Rethinking Benchmark and Contamination for Language Models with Rephrased Samples." arXiv preprint arXiv:2311.04850 (2023).

## Importance of reasoning tasks and the flexibility of DyVal to support more tasks

- First of all, why focusing on reasoning tasks in the paper? Reasoning ability is widely recognized as the core of both human and AI. Our focus on constructing reasoning tasks mirrors the intricate and multi-step nature of human reasoning [1, 2, 3], building reasoning benchmarks is a critical step to help LLMs towards intelligence.
- But our DyVal framework can construct tasks *beyond* reasoning tasks. Specifically, the DAGs in DyVal can also be applied to other tasks such as natural language understanding (NLU). Take sentiment analysis as an example, We can abstract a sentence into a syntax tree and form tree templates which can be used to generate diverse sentences. Take 'I am happy today' as an instance, it can be formalized as:

```
     Subject (I)
         |
     Predicate
    /        \
Verb (am)   Complement
            /        \
  Adjective (happy)  Temporal Phrase (today)
```



Following this template, we can create variations by replacing the corresponding words, like 'He was sad yesterday', 'She is excited now'.

[1] Brody, Nathan. "What is intelligence?." International Review of Psychiatry 11.1 (1999): 19-25.

[2] Lohman, D., & Lakin, J. (2011). Intelligence and Reasoning. In R. Sternberg & S. Kaufman (Eds.), The Cambridge Handbook of Intelligence (Cambridge Handbooks in Psychology, pp. 419-441).

[2] Sawada, Tomohiro, et al. "Arb: Advanced reasoning benchmark for large language models." arXiv preprint arXiv:2307.13692 (2023).


## We adopt graph structure, but not a paper on graph learning

We see that reviewers might mix our work with graph learning (GNN etc.) since our title contains the keyword "graph" and we adopt graph as the main method. But we would like to mention that we are not a graph learning paper, but a paper on large language models evaluation.

---

> ### Author Response · Authors · 2023-11-17
> **General response on paper revision**
>
> We thank all reviewers for their valuable comments in making this paper more comprehensive. We have updated the manuscript in the following aspects:
> - **Title**. We changed the title from "*DyVal: graph-informed dynamic evaluation of large language models*" to "*DyVal: dynamic evaluation of large language models for reasoning tasks*" as suggested by reviewer sT7J. This change is to emphasize the flexibility of our framework and our current focus on reasoning tasks.
> - **Related work**. We added more related work for a better explanation of the importance of the reasoning tasks in LLM evaluation. Then, we cited more recent efforts in LLM evaluation, especially data contamination. Additionally, we cited all the related work suggested by reviewers such as GraphWorld for a more comprehensive discussion.
> - More analysis.
> 	- **Flexibility**. As suggested by some reviewers, we further analyzed the flexibility of DyVal by extending it to natural language tasks. We added such experiments on sentiment analysis in Appendix H.
> 	- **Bias or imbalance analysis**. While our work naturally supports unbiased and balanced generation, we thank all reviewers for pointing this out. So, we added discussions to clearly state our DyVal naturally supports unbiased and balanced generation in Appendix F.
> - New experiments.
> 	- **Performance on general tasks after fine-tuning**. We added such experiments to discuss the performance of LLMs after fine-tuning on DyVal-generated samples for a better understanding in Appendix G.
>
> - - -
>
> We really hope that you can like this revision and give us more support. If you have more questions or concerns, please let us know:)
>
> Thanks
>
> Authors of DyVal

---

### Meta-Review · Area_Chair_pUMT · 2023-12-20

**Metareview:**

This paper proposes a method to evaluate the reasoning capability of LLMs. The key idea is to generate evaluation samples from a graph at certain complexity (size).  This would avoid the data contamination issue in prior LLM benchmarks. One standing out limitation is that this method requires manually specifying a problem as a computation graph with valid constraints.
The authors may want to elaborate how to ensure fair comparison with randomly generated evaluation samples.

**Justification For Why Not Higher Score:**

The paper is limited to reasoning tasks. The other limitation is the manually specifying a problem as a computation graph with valid constraints.

**Justification For Why Not Lower Score:**

The evaluation for LLM is very important. This paper proposes a method to avoid issues with a static benchmark.

---

### Decision · Program_Chairs · 2024-01-16

Accept (spotlight)